# A vegetable fat-based diet delays psychomotor and cognitive development compared with maternal dairy fat intake in infant gray mouse lemurs

Yohann Chaudron [1] ✉, Constance Boyer[2], Corinne Marmonier[2], Mélanie Plourde [3,4], Annick Vachon[3], Bernadette Delplanque[5], Mohammed Taouis [5] & Fabien Pifferi [1] ✉

Dairy fat has a unique lipid profile; it is rich in short- and medium-chain saturated fatty acids that induce ketone production and has a balanced ω6/ω3 ratio that promotes cognitive development in early life. Moreover, the high consumption of vegetable oils in pregnant and lactating women raises concerns regarding the quality of lipids provided to offspring. Here, we investigate maternal dairy fat intake during gestation and lactation in a highly valuable primate model for infant nutritional studies, the gray mouse lemur (*Microcebus murinus*). Two experimental diets are provided to gestant mouse lemurs: a dairy fat-based (DF) or vegetable fat-based diet (VF). The psychomotor performance of neonates is tested during their first 30 days. Across all tasks, we observe more successful neonates born to mothers fed a DF diet. A greater rate of falls is observed in 8-day-old VF neonates, which is associated with delayed psychomotor development. Our findings suggest the potential benefits of lipids originating from a lactovegetarian diet compared with those originating from a vegan diet for the psychomotor development of neonates.

During gestation and lactation, the mother must provide energy and specific nutrients to the neonate for brain development. In humans, the neonate's brain requires 74% of the total energy intake, although its mass is only 11% of that of the body[1]. Approximately 25% to 50% of its energy is provided by ketones as an alternative fuel to glucose[2,3]. Ketones are also the main source for lipid synthesis in the brain (saturated fatty acids and cholesterol) and thus meet the anabolic needs of the developing brain[4]. Ketones originate from short- and medium-chain fatty acid (SC/MC-FA) oxidation in the neonatal liver. In addition, other essential nutrients are needed for brain development, and one of these nutrients is docosahexaenoic acid (DHA; 22:6, n-3). This fatty acid is an omega-3 long-chain polyunsaturated fatty acid (n-3 LC-PUFA)[5] and is highly concentrated in the brain. At the neuronal level, DHA is thought to promote synaptic transmission, accounting for its possible function in memory and learning[6–8]. These abovementioned critical nutrients are delivered to offspring during gestation through placental transfer of DHA and SC/MC-FAs and during lactation through breastmilk. However, breastmilk DHA concentrations vary greatly across populations, with the highest concentrations in coastal populations (0.93% in Nigerian women)[9] and the lowest concentrations in vegan women (0.14%)[10]. Besides, linoleic acid (LA; 18:2, n-6) has increased during the last 60 years in the breastmilk of western populations (from ~5% to ~15%), whereas α-linolenic acid (ALA; 18:3, n-3) has remained unchanged[11]. These findings raise several concerns for two reasons: 1) DHA is synthesized from ALA by the enzyme Δ6-desaturase; 2) LA is also a substrate of the Δ6-desaturase and thus competes with ALA for access to the enzyme. Thus, this 3-fold increase in the LA:ALA ratio may be, among other factors, responsible for the low DHA supply to the neonate[12,13]. These modifications have been attributed to the drastic changes in lipid consumption in western populations following the Seven Countries Study[14], i.e., the decrease in animal fat intake (particularly butter) and the concomitant increase in vegetable oils rich in LA and palmitic acid (sunflower and palm oils). Since then, different manipulations of the maternal diet, such as DHA or marine food supplementation, have been used to restore the balance of the fatty acid supply of offspring.

[1]UMR CNRS MNHN 7179, 1 avenue du Petit Château, 91800 Brunoy, France. [2]Centre national interprofessionnel de l'économie laitière, 42 rue de Châteaudun, 75314 Paris cedex 09, France. [3]Centre de Recherche sur le Vieillissement, CIUSSS de l'Estrie – CHUS, 1036 Belvédère sud, Sherbrooke J1H 4C4, Canada. [4]Département de Médecine, Université de Sherbrooke, Sherbrooke, Canada. [5]UMR 9197, Paris-Saclay Institute of Neurosciences (NeuroPSI), University of Paris-Saclay, CNRS, 151 route de la Rotonde, F-91400 Saclay, France. ✉e-mail: yohann.chaudron@mnhn.fr; fabien.pifferi@mnhn.fr

However, maternal DHA supplementation often fails to demonstrate an association with infants' cognitive scores, as depicted by successive meta-analyses and systematic reviews[15–17]. Aside from these debated impacts on cognitive scores, DHA supplementation has rarely been implemented for motor development testing. In 2009, a randomized controlled trial demonstrated that daily supplementation with 20 mg DHA throughout the first year of life resulted in early achievement of early motor milestones (such as reaching for an object to touch, bringing toys to the mouth, or sitting without support), although this was not the case for later motor milestones (i.e., standing alone, walking alone)[18]. Since then, associations between DHA and motor development have been controversial, with most studies showing no association[19–21] or only if DHA intake originated from marine foods[22]. Several hypotheses could explain this lack of evidence: (1) Motor functions are tested relatively late regarding the milestones of motor development (between 12 and 18 months), whereas Agostoni and colleagues suggested that only early milestones might be affected by supplementation[18]. This is again supported by a prospective cohort study that revealed an association between milk DHA and motor skills at 6 months[23]. (2) DHA supplementation is usually based on a single-nutrient approach, concealing a wide variety of complexes and matrices in which DHA may settle in or interact with other nutrients inside foods. A food approach might then be needed instead of a single-nutrient approach.

Dairy fat has been poorly investigated but has been identified as an interesting nutrient source for neonatal brain development. Dairy fat is particularly rich in SC/MC-FAs, with levels ranging from 20% to 29%[24]; these levels might provide large amounts of ketones for brain development. Although dairy fat contains low levels of DHA (0.01% of total fatty acids), dairy fat displays an optimal LA:ALA ratio between 2 and 3, which is favorable for ALA bioconversion to DHA[25] and meets the French nutritional guidelines[26]. Finally, dairy fat provides important amounts of vitamins A and D, which are also brain-selective nutrients[5,27]. Consequently, dairy fat may be able to address the challenges of brain development. Dairy fat supplementation has only been conducted in rodent models, and an increase in brain DHA levels has been demonstrated[25,28]. However, the cognitive impacts have only been assessed in an adverse situation after the infusion of lipopolysaccharide[29]. Indeed, rats born to dams fed dairy fat have been shown to maintain normal working memory, whereas this was not the case for rats born to dams fed vegetable oils. With the increasing popularity of vegetable oils in the mother's diet and the concerns that arise about the development of the offspring's brain, the use of a nonhuman primate model in a nonadverse situation would be more accurate for extrapolating potential health benefits for human infants. Specifically, observational studies on this topic in humans have been more than controversial, probably due to the confusion between the different dairy products that display different nutritional properties and matrix organization, necessitating additional interventional studies[30–32]. In this regard, how maternal dairy fat intake during gestation and lactation might impact the cognitive performance of neonates in primates constitutes a critical gap in the literature.

Mouse lemurs (*Microcebus murinus*) are nocturnal arboreal lemuri-form primates originating from Madagascar that exhibit several interesting features for infant nutritional studies. They share phylogenetic and physiological proximity with humans, as demonstrated by genomic studies[33]. Mouse lemurs are omnivorous animals that eat insects, fruits, vegetables, and even meat, hence mimicking the human diet to some extent. Their size (~15 cm and 60–80 g), light-inducible highly prolific reproduction capacity for primates (2 pups per mother and per year), and rapid development (2 months of gestation and 2 months of weaning) make them easy to manipulate and breed in captivity[34]. Despite the major interest in aging biology[35], experimental data related to the early psychomotor development of the gray mouse lemur are scarce. Two experimental diets, in which the lipid fractions are either based on dairy fat (DF) or vegetable fat (VF), were designed and provided to female mouse lemurs two months before conception, during gestation, and until weaning. The VF diet, although within the recommendations for energetic content, is thought to be a reflection of the western diet insofar as most of its fatty acids are derived from vegetable oils (olive, sunflower, rapeseed, and palm) and thus constitutes our control condition. In contrast, the DF diet is a modification of the VF diet, with 75% of the vegetable lipids replaced with dairy lipids. Using a new battery of tests specifically developed for infant mouse lemurs, we showed that neonates from mothers fed a dairy fat-based diet displayed improved psychomotor performance, most likely due to increased motor coordination, sensory development, and/or memory abilities. Motor coordination differences between mouse lemurs fed a DF versus VF diet increase from 8 days postpartum and progressively decline. This phenomenon is in contrast with memory abilities, which still differ at 30 days postpartum.

## Results

### Offspring characteristics

Neonates weighed 15.2 ± 0.5 g at D + 8 and reached 37.1 ± 0.7 g at D + 30 (Fig. 1). The VF and DF groups exhibited similar body weights at D + 8 and D + 15. At D + 22 and D + 30, the VF group had a significantly greater body weight than did the DF group (mixed linear model, post hoc comparisons at D + 22 and D + 30, $t(189) = 2.06$, $\hat{\beta} = 2.75$ [0.11 – 5.39], p = 0.041 and $t(189) = 2.09$, $\hat{\beta} = 2.79$ [0.15 – 5.73], p = 0.038, respectively). Body weight did not differ during the first 5 days of life (Supplementary Table 1).

Despite the differences in body weight from D + 22, this did not affect performance (see following sections). Weight differences between dietary groups were mainly explained by litter size. Indeed, the body weight of single neonates at D + 30 was significantly greater than that of neonates belonging to larger litter sizes of 2 and 3 pups (mixed linear model, post hoc comparison, $t(45) = 2.76$, $\hat{\beta} = 6.14$ [0.75–11.52], p = 0.022 and $t(45) = 2.09$, $\hat{\beta} = 7.03$ [2.49–11.58], p = 0.001, respectively; Fig. 2). The distribution of litter size according to diet was similar at birth. However, after the death of one VF pup in two twin litters, single neonates became more common in VF animals (5 vs. 2). Of note, the small sample sizes did not allow statistical comparison at this point (Supplementary Table 2). We did not find a bias in the sex ratio according to the diet group (Supplementary Table 2).

### Negative geotaxis (day+8)

Regarding the distribution of success, DF animals were more than twice as likely to succeed in this task than VF animals, as depicted by their greater proportion of success (54.67% vs. 24%, mixed beta-regression, z = 6.25, OR = 0.41 [0.20 – 0.87], p = 0.019; Fig. 3a). As a corollary, VF animals were more likely to fall off the device. In total, 22 VF animals (~88%) fell at least once, whereas only 12 (~48%) DF animals fell (mixed model with a binomial distribution, z = 2.41, OR = 16.85 [1.69–168.29], p = 0.016; Fig. 3b). After exclusion of all animals that failed this task, i.e., animals that fell or did not reach the mark within the time frame (exact number in Supplementary Table 3), we did not find any effect of diet, sex or weight on the time to reach the horizontal and vertical marks.

### Horizontal rod: early psychomotricity (day+8)

Across all levels of difficulty, i.e., at 11 cm, 22 cm, and 33 cm from the nest box, the number of DF animals that succeeded the horizontal rod task was significantly greater than that of VF animals (mixed model with a binomial distribution, z = −3.05, OR = 0.11 [0.03 – 0.46], p = 0.002) (Fig. 4a). At the easiest level, 92% of the DF animals succeeded, whereas 60.9% of the VF animals succeeded. The percentage of successful VF animals decreased with increasing difficulty (from 60.9% to 40.9%), whereas 79.2% of the DF animals succeeded at the hardest level (Fig. 4a). Furthermore, falls did not differ across levels for either diet; however, they were much more frequent in VF animals (~20.3% vs. ~1.3% for DF animals) (mixed model with a binomial distribution, z = 2.63, OR = 20.64 [2.17 – 196.48], p = 0.008). Regarding locomotor strategy, successful animals were more likely to never exhibit helicoid movements, independent of their diet (mixed model with a binomial distribution, z = −2.85, OR = 0.04 [0.01 – 0.38], p = 0.004).

After excluding all animals that failed this task, i.e., animals that fell or did not reach the nest box (exact number in Supplementary Table 4), we did not find any difference between the DF and VF animals despite a modest degree of evidence for a decreased time to reach the nest box for the DF

**Fig. 1 | Body weight of mouse lemur neonates according to maternal diet from D + 8 to D + 30.** Statistical tests were performed using a mixed linear model. *: $p < 0.05$, $n = 25$ for each diet group. Box plot elements: the centerline is the median, the box limits are the upper and lower quartiles, and the whiskers represent 1.5-fold the interquartile range.

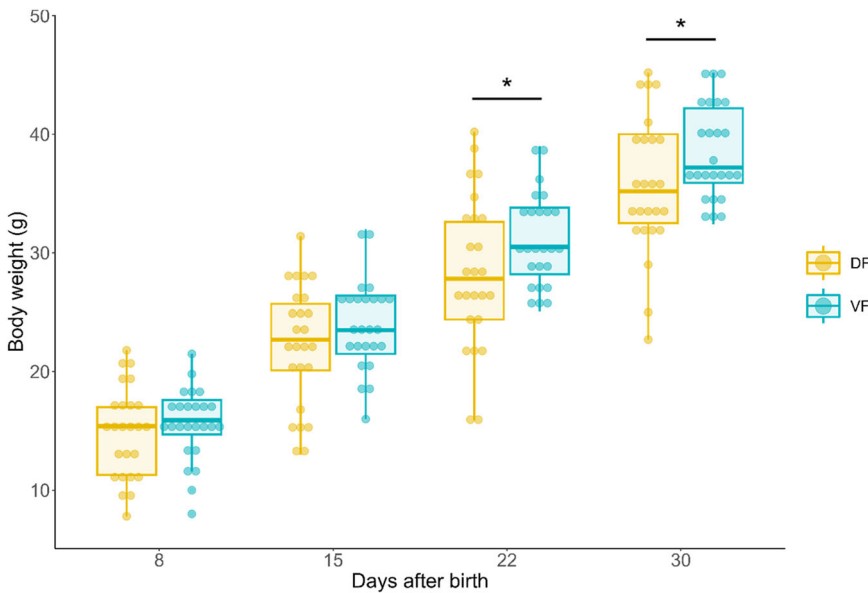

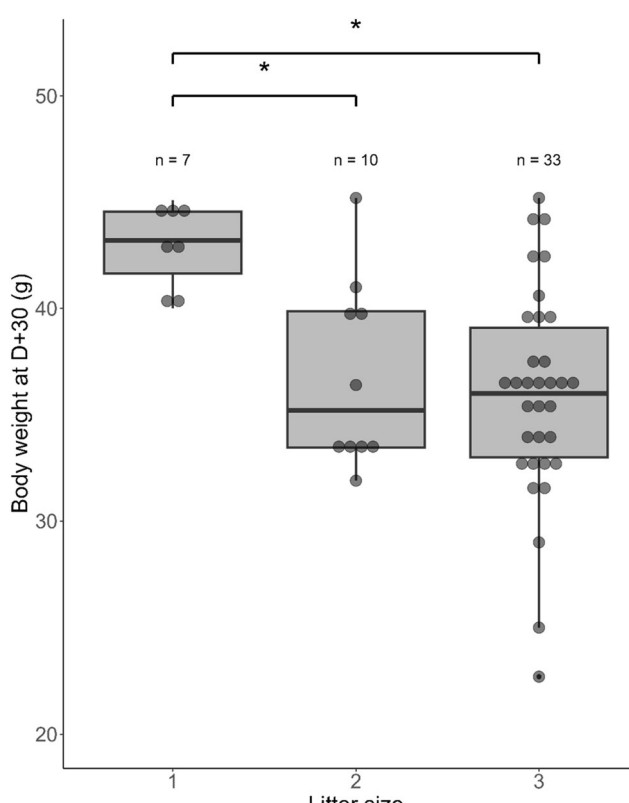

**Fig. 2 | Body weight of mouse lemur neonates at D + 30 according to litter size.** Statistical tests were performed using a mixed linear model. *: $p < 0.05$, **: $p < 0.01$; n indicates the number of animals. Box plot elements: the centerline is the median, the box limits are the upper and lower quartiles, and the whiskers represent 1.5-fold the interquartile range.

animals (mixed model with a gamma distribution and a log link function, $z = 1.94$, $\hat{\beta} = 66.41$ [47.74–92.39], $p = 0.053$) (Fig. 4b). Post hoc comparisons revealed no significant differences at each level of difficulty between the DF and VF groups, despite again showing a modest degree of evidence for a decreased time to reach the nest box in DF animals only for the first level of difficulty, i.e., at 11 cm from the nest box ($p = 0.053$). We did not find any effect of sex or weight on the time to reach the nest box for any level of difficulty. The time to reach the nest box did not differ between levels of difficulty. The latency of the first movement also decreased between levels 1 and 2 (mixed model with a gamma distribution and a log link function, $z = -2.00$, $\hat{\beta} = 0.47$ [0.22–0.98], $p = 0.045$) but did not differ between the two dietary groups. The number of DF animals that did not exhibit helicoid movements was significantly greater than that of VF animals (mixed model with a binomial distribution, $z = 2.40$, OR = 51.11 [2.05–1273.54], $p = 0.016$) and remained unchanged across levels of difficulty. Animals expressed fewer and fewer helicoid movements by cm when the level of difficulty increased (beta-regression, 11 cm vs. 33 cm, comparison, $z = -2.63$, OR = 0.49 [0.29–0.83], $p = 0.009$).

### Rotating rod: late psychomotricity (day+15, +22 and +30)
The maximum time spent on the rod increased six-fold from D + 15 to D + 30 (mixed model with a gamma distribution and a log link function, $z = 6.72$, $\hat{\beta} = 5.43$ [3.32 – 8.90], $p < 0.001$) (Fig. 5). The variance also increased. The scores at D + 15 were highly reproducible, whereas the scores at D + 30 exhibited high variability (Levene test, $F(3,193) = 16.9$, $p < 0.001$). Globally, DF animals stayed on the rod longer (mixed model with a gamma distribution and a log link function, $z = -4.52$, $\hat{\beta} = 0.52$ [0.39 – 0.69], $p < 0.001$). Post hoc tests confirmed differences at D + 15 and D + 22 but not D + 30 ($p < 0.001$, $p = 0.017$ and $p = 0.517$, respectively). Body weight and day postpartum at which tasks occurred were positively correlated (mixed linear model, $p < 0.001$; Fig. 1), so the effect of body weight on time spent on the rod was analyzed within groups of the same age. However, consistent across the different age groups, body weight had no effect on time spent on the rod. Regarding the task outcomes, independent of their diet, animals fell in 52.4% of the trials at D + 15, and this percentage significantly decreased as the animals grew (mixed model with a negative binomial distribution on the number of falling trials, $z = -4.85$, $\hat{\beta} = 0.10$ [0.04–0.26], $p < 0.001$) (Fig. 6a). Over time, DF animals fell ~20% less than VF animals (mixed model with a negative binomial distribution, $z = 1.98$, $\hat{\beta} = 1.67$ [1.01–2.78], $p = 0.047$). At D + 30, the DF animals fell during only ~4% of the trials, and the VF animals still fell 25% of the time (post hoc test, $p < 0.001$). Jumps also tended to increase over time (mixed model with a negative binomial distribution, $z = 1.93$, $\hat{\beta} = 2.13$ [0.99–4.59], $p = 0.053$), irrespective of diet (Fig. 6c). DF animals hung on for 3 rotations more frequently than VF animals did (mixed model with a negative binomial distribution, $z = -2.02$, $\hat{\beta} = 0.49$ [0.25–0.98], $p = 0.043$; Fig. 6b); however, this behavior did not differ across time. Finally, refusals did not differ across time or diet (Fig. 6d).

**Fig. 3 | Negative geotaxis results (D + 8). a** Violin plot showing the probability density of the number of successes per animal (ranging from 0 to 3 out of 3 trials). Each point represents an animal, and the crossbar represents the mean. Statistical tests were performed using a GLM with a beta-regression. **b** Number of animals that fell at least once on the negative geotaxis task. Statistical tests were performed using a GLM with a binomial distribution. *: $p < 0.05$, **: $p < 0.01$.

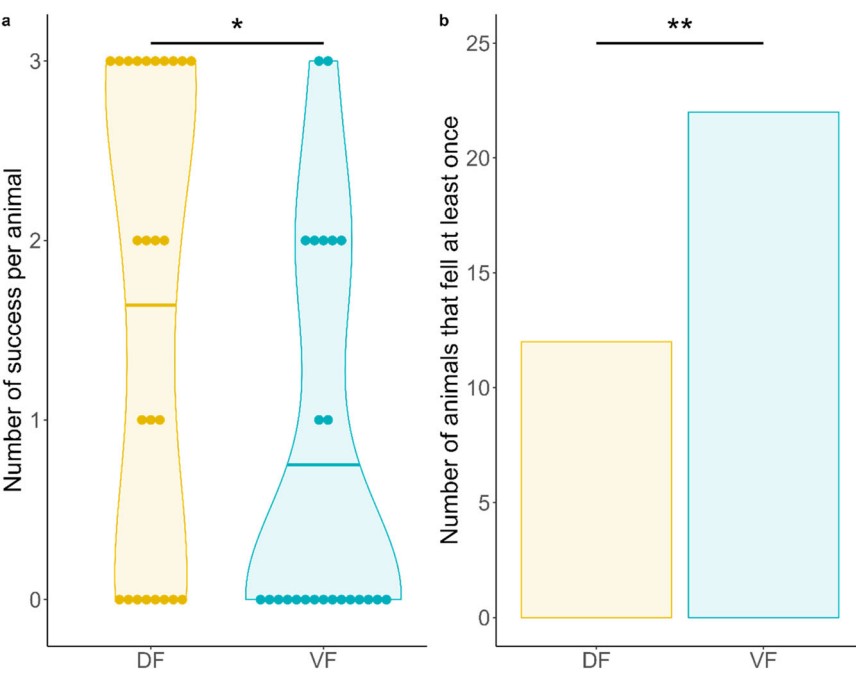

## Visual discrimination task in the open field (day+15 and +30)

The time to reach the nest box in the open field decreased across trials at D + 15 and D + 30 (Fig. 7). Because the animals that failed were assigned a score of 300 sec, the decrease in time to reach the nest box was mainly driven by a decrease in the number of animals that failed (mixed model with a zero-inflated gamma distribution and a log link function on the variable "300-time", z = −3.94, $\hat{\beta} = 0.66$ [0.53–0.81], $p < 0.001$ for the zero-inflated model, z = 1.44, $\hat{\beta} = 1.04$ [0.98–1.11], $p = 0.15$ for the conditional model) (Fig. 8a). The number of successful animals was not affected by diet, despite a modest degree of evidence for a stronger increase in successful DF animals across trials (mixed model with a zero-inflated gamma distribution and a log link function on the variable "300-time", z = 1.93, $\hat{\beta} = 1.32$ [1.00–1.76], $p = 0.054$ for diet:trial interaction in the zero-inflated model). Post hoc comparisons revealed a greater number of successful DF animals than successful VF animals for the fourth trial (15 vs. 8, p = 0.030) and the fifth trial (19 vs. 10, p = 0.004). Among successful animals, new learners (i.e., animals that succeeded for the first time) and recalling animals (i.e., the proportion of successful animals among those that had already learned) were listed for each trial. The number of new learners did not differ between diet groups, and the total number of animals that learned at least once at the end of the six trials did not differ (Fig. 8a). The proportion of recalling animals was relatively stable across trials and ranged from 69.2% to 76%, with the exception of the fourth trial, wherein it decreased to 45.2%. This decrease was significant for the VF group (mixed model with a binomial distribution, post hoc comparison, trial 3 vs. trial 4: −42.2%, p = 0.048), whereas it was not significant for the DF group (−19.2%, p = 0.238). During the following trial (5th), there was a modest degree of evidence for a greater proportion of recalling DF animals than recalling VF animals (mixed model with a binomial distribution, post hoc comparison, 81.8% vs. 50%, p = 0.056) (Fig. 8b).

## Discussion

Our data suggest that providing mothers a DF diet improved cognitive and psychomotor scores in newborns during their first 30 days of life. This pattern is congruent across all tasks and may involve several functions, such as motivation, strength, motor coordination, sensory perception, learning and even memory. The levels of anxiety and motivation in mouse lemurs can be modified by diet[36,37] and hence impact performance. Increased anxiety and/or lack of motivation are generally inferred from a greater

latency to first movement[38,39]. However, the latency to first movement during the horizontal rod task did not differ according to diet in our study. Animals from both dietary groups began to walk sooner and reached the nest box faster when difficulty increased. These data suggest that motivation increased, or anxiety decreased, with increasing difficulty, similar to that of a habituation process. In addition, crossing the horizontal rod might also involve substantial muscular strength. According to our data, the maximum time spent on the rotating rod was associated with an increased frequency of hanging on for 3 rotations, which may be related to increased grasping and strength abilities. Very young mouse lemurs have been shown to exhibit hand pull strength similar to that of adult lemurs, relative to their body weight, as soon as 8 days of age[40]. Consequently, the increase in pull strength is dependent on body mass gain. However, we did not find any differences in body weight according to diet at D + 8, suggesting that performance on the horizontal rod is unlikely to be associated exclusively with strength. In contrast, VF animals had a greater body weight at D + 22 but stayed unexpectedly shorter on the rotating rod than did DF animals, suggesting that diet had minimal impact on strength but rather a notable effect on fat mass. However, we did not record this outcome. Instead of strength, falls in VF animals might have been mediated by sensory deficits. Indeed, negative geotaxis has been widely used to assess graviception in other arboreal primates, such as slow lorises and greater galagos[41], squirrel monkeys[42] or common marmosets[43], showing that all animals displayed negative geotaxis during the first days or weeks of life. This behavior is then acquired very quickly, underlining its potential importance for balance functioning in arboreal animals that progress at great heights. This is also the case for mouse lemurs. Graviception depends on the vestibular system, which is morphologically well developed at birth in mammals[44], but its maturation occurs during the first few months postpartum due to the sensory experience of gravity[45]. Hence, given that more than half of our animals did not succeed in the negative geotaxis task, we hypothesize that the maturation of vestibular function was still an ongoing process at D + 8. Nevertheless, the mean success rate for DF neonates was twice as high as that for VF neonates. It is thus likely that a DF diet might have interfered with the onset of vestibular maturation. The remaining differences in performance between dietary groups among successful animals suggest that, in addition to sensory development, fine motor coordination might be involved. Indeed, the number of successful DF animals that never exhibited helicoid movements in the horizontal rod task was greater than that of successful VF animals.

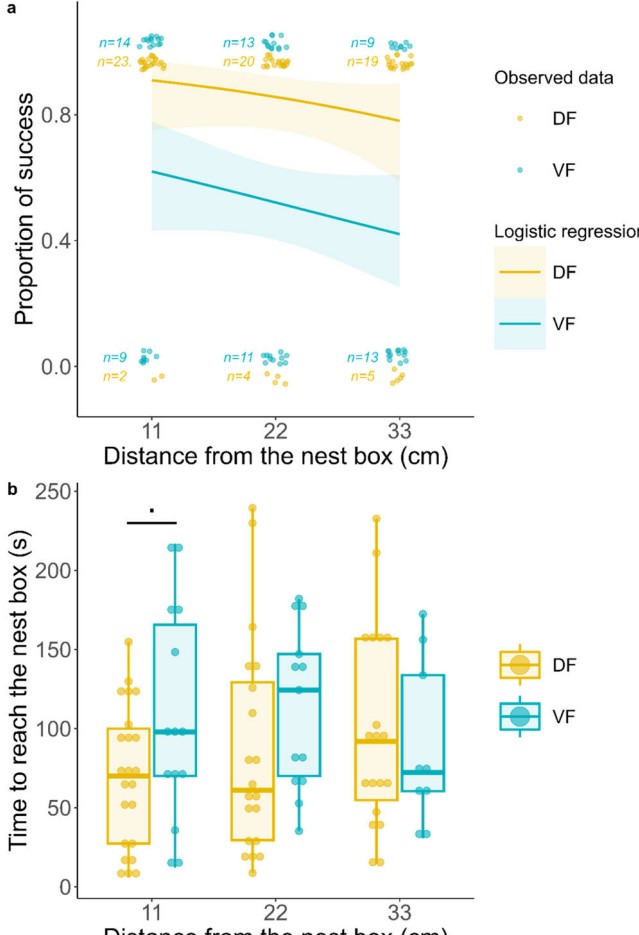

**Fig. 4 | Horizontal rod test results (D + 8). a** Logistic regression modeling of the probability of success according to the distance from the nest box and the diet. Each animal, depicted by a point, is successful (assigned a value of 1) or fails (assigned a value of 0). The curve and its 95% confidence interval are generated using a mixed generalized model with a binomial distribution. The VF curve differed significantly from the DF curve ($p = 0.002$). **b** Box plot of the time to reach the nest box among successful animals for each distance. The number of successful animals for each distance level and diet group is provided in Supplementary Table 4. Statistical tests were performed using a GLMM with a gamma distribution and a log link function. ·: $p < 0.10$ (trend). Box plot elements: the centreline is the median, the box limits are the upper and lower quartiles, and the whiskers represent 1.5-fold the inter-quartile range.

Helicoid movements are generally associated with failures and falls and could therefore be a proxy of delayed motor coordination development. Indeed, the acquisition of crawling and climbing typically occurs between D + 6 and D + 8 and is delayed to D + 14 in hand-reared animals, which typically lack maternal milk[46]. Therefore, DF animals do seem to fit quite well with expectations, and VF animals might suffer from delayed acquisition of these early motor milestones. In contrast, in DF animals, helicoid movements are supplanted by symmetrical lateral walking, which is the most time-efficient strategy. This sequence has already been observed in adult mouse lemurs when walking at a very low speed, which is the case for neonates[47,48]. Late acquisition of this strategy in VF animals may be explained by a late grasping posture transitioning from a schizaxonic posture to a mesaxonic posture[40], although it remains poorly understood how these postures might influence the stability or locomotion on horizontal rods. The overall decrease in the number of helicoid movements by cm when difficulty increased could be explained by psychomotor learning during the task, which was independent of diet. The homogenous ability to first associate the stimulus and the reward in the visual discrimination task

suggested that cognitive learning was not affected by diet. This finding also suggested that 3-dimensional vision was acquired equally, as would have been expected at this age since parallax head movements usually occur for the first time between D + 10 and D + 13[46]. Nevertheless, animals could also use olfactory cues to find their nest box given that mouse lemurs, similar to other Strepsirrhines, rely greatly on olfaction to navigate and communicate[49,50]. However, the smaller decrease in the number of DF-recalling animals in the fourth trial, which occurred 15 days after the third trial, indicates that DF may sustain long-term memory instead of learning. Overall, we propose that the DF diet might have positively affected graviception, motor coordination and memory abilities. However, exclusively assessing successful animals, differences between dietary groups are rare. We suggest that diet may have impacted the onset of psycho-sensory-motor development but not the acquisition of this development. The decline in the differences of time spent on the rotating rod between dietary groups from D + 15 to D + 30 suggests that once the main psychomotor functions are acquired, diet may no longer influence performance. However, whether DF animals exhibit earlier development or VF animals exhibit delayed development is uncertain. A study in the same colony showed that infant mouse lemurs reached less than 5% of the maximum score (300 s) on the rotarod at D + 15, which makes them quite similar to our VF animals[40]. Their DF counterparts reached 12.8% of the maximum score, suggesting that DF animals may have actually experienced earlier psychomotor development. This pattern is consistent with what has been found in several DHA supplementation studies, with most differences observed before or around the sixth month (e.g., reaching for an object to touch at 16.9 weeks vs. 18.0, bringing a toy to the mouth at 17.7 weeks vs. 18.6, and sitting without support at 26.1 weeks vs. 27.2)[18] but not after. The first six months of life in humans are exactly the period of exclusive breastfeeding[51], which corresponds to the first 15 days of life in mouse lemurs. Thus, the decrease in differential motor development due to n-3 PUFA intake after exclusive breastfeeding might constitute a recurrent pattern.

Nevertheless, we must acknowledge that diet is far from being the only factor potentially involved in psychomotor development. Maternal investment can be assessed using gestation length and offspring body mass. The gestation length did not differ between the dietary groups and was consistent with the literature (i.e., $61.5 \pm 0.9$ days[52]). In early life, maternal investment per neonate is known to differ according to parity, litter size, and the presence of male neonates. Indeed, multiparous females display more close contact with their offspring, which may result from experience-enhanced maternal behavior[52]. Given that only multiparous females were used for this study, the reproductive maternal experience should not constitute a cofounding factor. In addition, single neonates benefit more from maternal care than their counterparts from larger litter sizes, with a daily gain in body mass of 1.12 g (vs. 0.97 g for triplets), which is consistent with previous studies[53,54]. Moreover, female neonates from unisex litters (i.e., composed of only females) demonstrated reduced growth and increased mortality[55]. We also found this relationship in our study, with triplets consisting only of females demonstrating a lower daily gain in body mass (1.01 g vs. 0.90 g for females originating from mixed litters). Mother mouse lemurs are also known to display greater time in close contact with male neonates or female neonates from mixed litters, suggesting a maternal preference for males[52]. Although litter size and sex might cause differential neonatal development, we did not observe any effect of these factors on psychomotor or cognitive outcomes. Variation in the sex ratio between dietary groups, although not statistically significant, might originate from the maternal social environment. Indeed, isolated females prior to conception preferentially give birth to females, whereas females grouped together preferentially give birth to males[56]. However, in our study, females were always grouped together according to their diet 2 months prior to conception. Thus, we would have expected a bias in the sex ratio toward males, which is not the case. We hypothesize that the only force at stake was random sampling, which potentially resulted in important sex ratio bias due to the small sample size. Unexpectedly, a greater number of single neonates at birth was observed compared to that previously reported (21.7% in our study vs. 9%[54]),

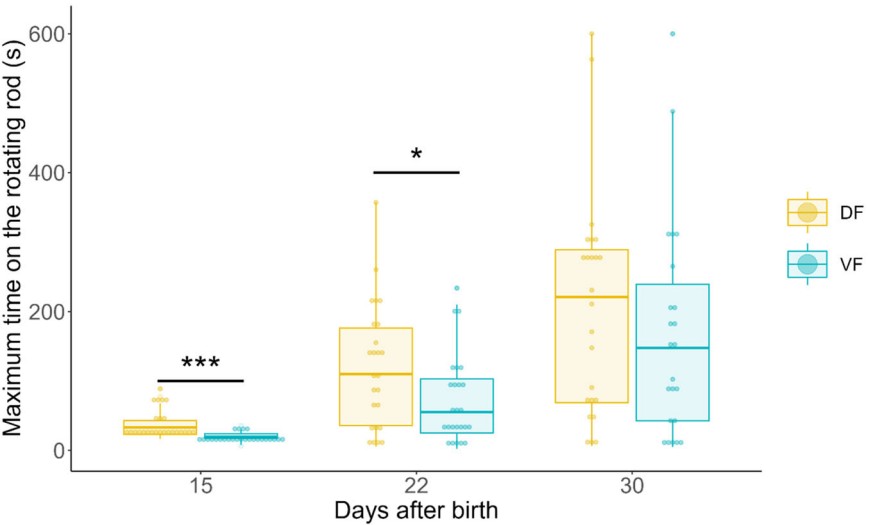

**Fig. 5 | Maximum time spent on the rotating rod at D + 15, D + 22 and D + 30.** Statistical tests were performed using a GLMM with a gamma distribution and a log link function. $*p < 0.05$, $***p < 0.001$. Box plot elements: the centreline is the median, the box limits are the upper and lower quartiles, and the whiskers represent 1.5-fold the interquartile range. $n = 25$ in each diet group at D + 15 and D + 22; $n = 24$ (DF) and n = 23 (VF) at D + 30.

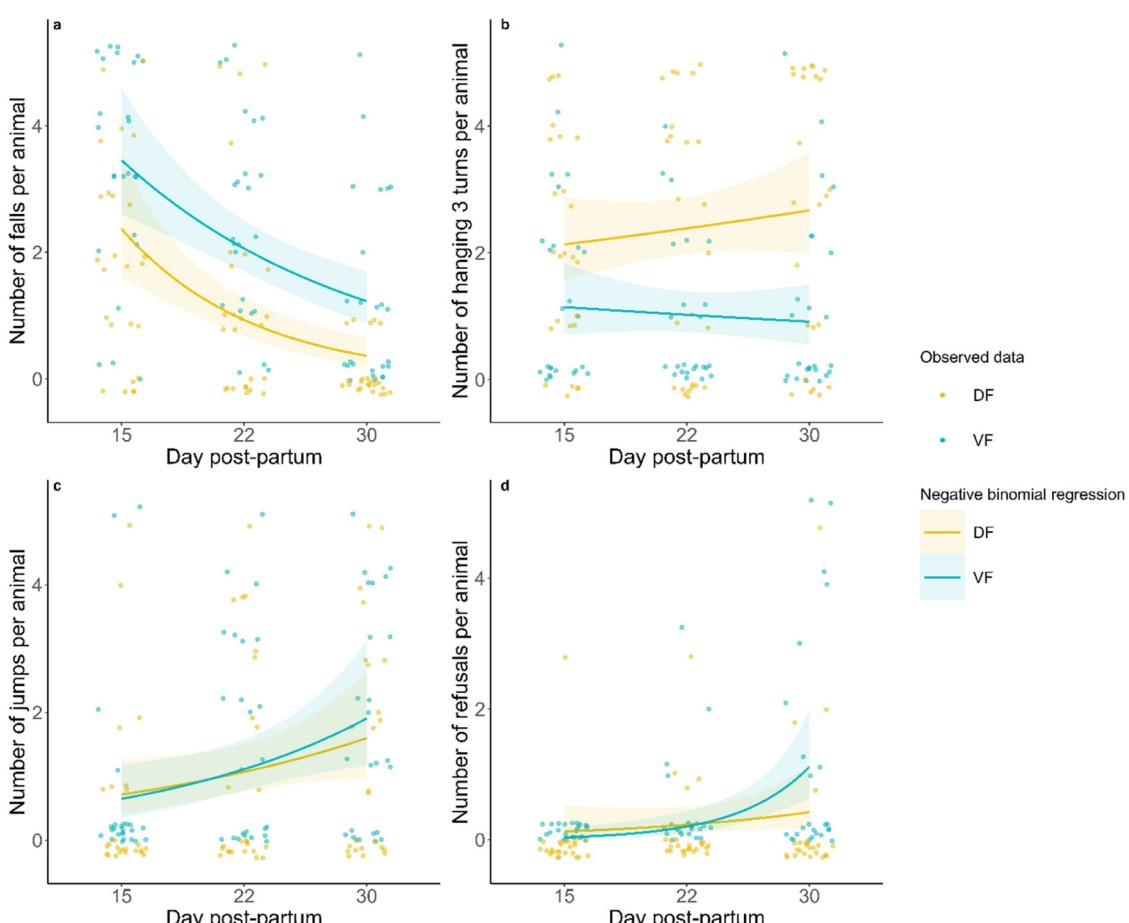

**Fig. 6 | Rotating rod outcomes at D + 15, D + 22 and D + 30.** Negative binomial regression modeling the number of each rotating rod outcome per animal, i.e., falls (**a**), hanging on for 3 turns (**b**), jumps (**c**) and refusals (**d**), at D + 15, D + 22 and D + 30 according to diet. Each animal, depicted by a point, can be assigned a value from 0 to 5 out of 5 trials for each outcome. The curve and its 95% confidence interval are generated using a mixed generalized model with a negative binomial distribution. The VF curve differed significantly from the DF curve only for 'falls' and 'hanging on for 3 turns' outcomes ($p = 0.025$ and $p = 0.046$, respectively). $n = 25$ in each diet group at D + 15; $n = 25$ (DF) and $n = 24$ (VF) at D + 22; $n = 24$ (DF) and $n = 25$ (VF) at D + 30.

explaining the greater mean body mass of our cohort (37 g at D + 30 in our study vs. 32 g[54]). Typically, triplets are more frequent (55%), followed by twins (36%) and singletons (9%). However, this might be explained by the young age of the mothers included in this study given that maternal age at conception has been shown to be positively associated with litter size[54]. Litter size at birth did not differ between dietary groups (Supplementary Table 2). However, because one neonate died in two twin litters in the VF group, heterogeneity in litter size appeared between the dietary groups, with the VF group displaying more single neonates than the DF group. This heterogeneity was reflected by the greater mean body mass of VF neonates at

D + 30, as expected. Last, although the diets were isocaloric, the dietary intervention could have had a direct effect on body mass gain. Indeed, SC/MC-FAs prevent obesity due to their preferential oxidation by the liver[57], whereas perinatal LA intake promotes adipogenesis in rodents[58–60] and humans[61]. Thus, it is not unlikely that the DF diet, which is rich in SC/MC-FAs and has a lower LA/ALA ratio, induced a lower body mass gain than the VF diet. Apart from maternal investment, the neonatal social environment, such as contact with litter mates, is also a potential confounding variable. For

instance, grooming behavior toward others begins around D + 7, and playing is initiated between D + 10 and D + 13[46]. Playing with others may take different forms, including play directed toward the mother ("follow-my-leader" game and pouncing on the mother's tail or hands) or litter mates (wrestling, mounting of biting)[62]. The latter playing form constitute a simulation of agonist behavior and/or copulation[53] and might help young mouse lemurs learn their motor abilities and understand their environment. However, there are no data regarding to what extent early social stimulation

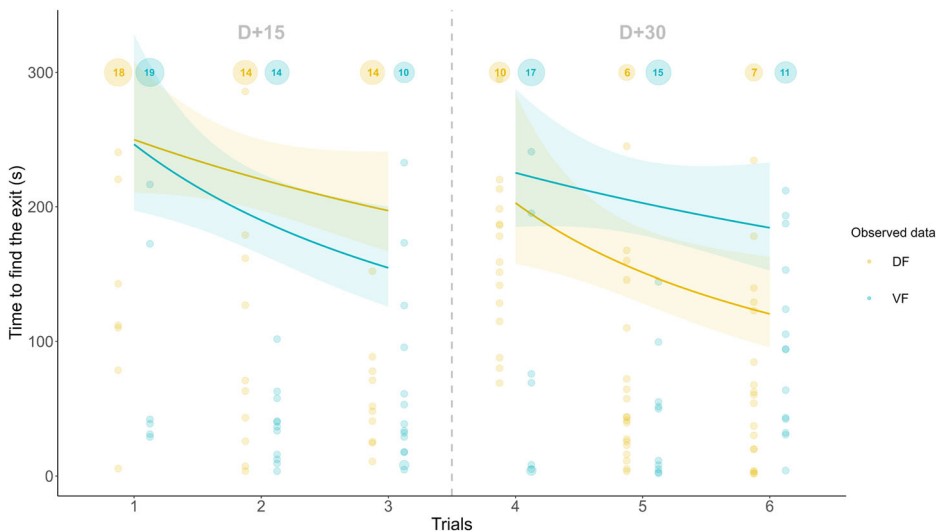

**Fig. 7 | Gamma regression modeling of the time to find the exit across trials at D + 15 and D + 30 for the visual discrimination task.** Each animal is depicted by a point. Animals that failed this task are assigned the maximum value (300 s). The over-lapping points inherited from these animals are melted in a larger dot proportional to their number, which is displayed inside. The curves and their 95% confidence intervals are built using mixed generalized models with a gamma distribution and a log link function. The DF and VF curves did not differ significantly from one another ($p = 0.436$ for D + 15 and $p = 0.305$ for D + 30). $n = 25$ in each diet group and for each trial.

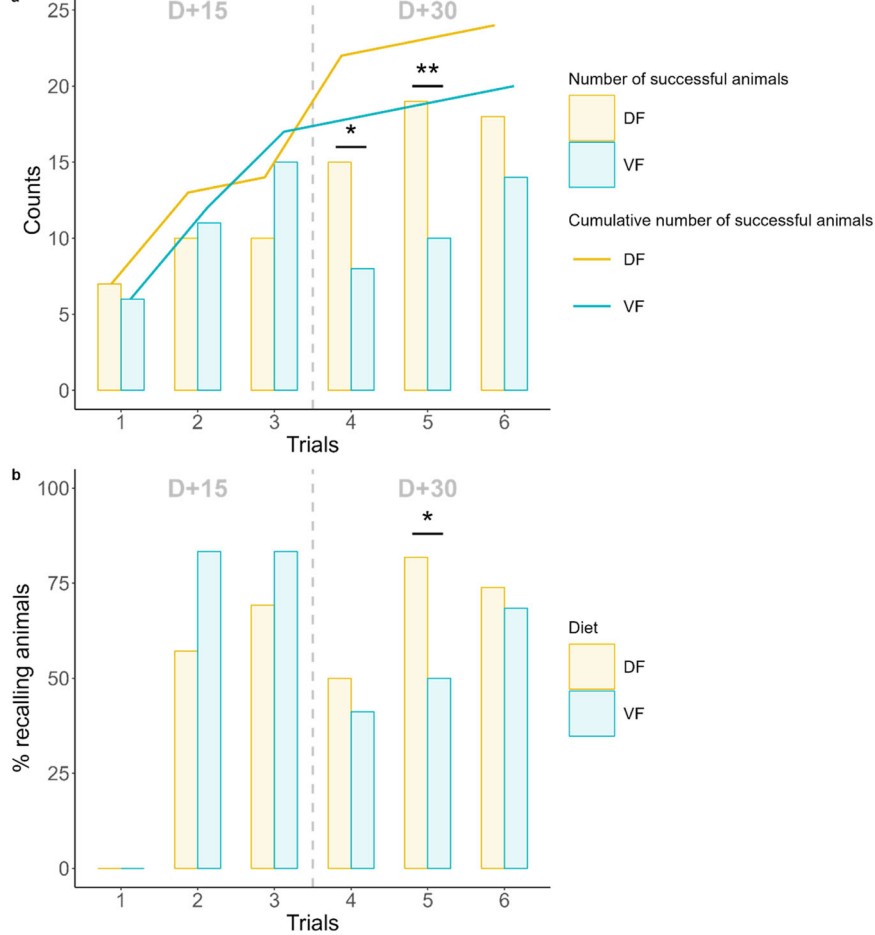

**Fig. 8 | Learning and memory abilities according to diet during the visual discrimination task. a** Number of successful animals in each trial for the visual discrimination task and cumulative curve of the number of animals having learned, i.e., having succeeded at least once, and (**b**) proportion of recalling animals, i.e., successful animals among those that had already learned during a previous trial, for the visual discrimination task. Statistical tests were performed using a mixed model with a binomial distribution. *: $p < 0.05$; **: $p < 0.01$.

might impact psychomotor and cognitive development in mouse lemurs. Overall, it cannot be excluded that differences in maternal investment and social environment between groups impacted our results in some manner in addition to dietary intervention.

All of these behavioral impacts on infants must originate from the nutrients ingested by the mother during gestation and lactation, at least until $D + 15$. Indeed, it has been reported that mouse lemur infants are always in the nest box until $D + 16$ and depend exclusively on maternal milk[52] because mothers do not carry and give solid food to their offspring[46]. There is no doubt that if the performance of individuals during the negative geotaxis and the horizontal rod tasks at $D + 8$ indeed depended on the diet, it was primarily dependent on the maternal diet. However, owing to ethical and methodological reasons, we did not sample milk or infant plasma for fatty acid composition, which is one major limitation of our study. Thus, we could only hypothesize about the pathways involved in the observed functional effects. Recent epidemiological data demonstrated that maternal dairy product consumption during pregnancy was associated with decreased LA in colostrum and increased DHA levels, and a decreased LC-PUFA ω6:ω3 ratio in cord red blood cells[63]. As expected, these data support the hypotheses that although dairy products are depleted of DHA, DF consumption may enable the maternal synthesis of DHA, which is then provided through the umbilical cord and maternal milk to the offspring. This synthesis may be surprising given the very limited bioconversion rate (< 0.2%) of ALA to DHA[64] and could be explained by the optimal LA/ALA ratio of DF. Thus, we expect that maternal DF uptake in mouse lemurs might influence the composition of milk and cord red blood cells by increasing DHA and myristic acid. In humans, dietary DHA is primarily transferred to the fetus from maternal blood for most of the pregnancy. Dietary DHA is even stored in maternal adipose tissue until the end of pregnancy ( ~ 30 weeks) when DHA is eventually released for fetal accretion at an extraordinary rate (40–57 g/day[65,66], with a 5- to 15-fold greater concentration in the fetal circulation than in the maternal circulation[67]). This bioaccumulation phenomenon puts the mother at risk of DHA depletion. Enhanced DHA synthesis from ALA might then support maternal DHA transfer by preventing it from being depleted at the mother's expense. In contrast, ALA and LA are less concentrated in the fetal circulation than in the maternal circulation[67]. Given that Δ6-desaturase activity is quite immature in the fetal liver[68], fetal DHA is more likely to be derived from maternal synthesis than fetal synthesis, although there are no data on Δ6-desaturase activity in mouse lemurs. Indeed, DHA synthesis has been shown to be enhanced during pregnancy potentially because of the constant high levels of estrogens[69–71]. MC-FAs (especially C10:0 and C12:0) have also been shown to preferentially bioaccumulate 2- to 3-fold in fetal circulation[67]. However, it seems unlikely that ketones play a direct role in the fetal brain at this step because it is estimated that ~80% of maternal ketones absorbed by the placenta are metabolized in situ and not transferred to the fetus[72]. Nonetheless, placental ketone metabolism allows more glucose to be available for fetal needs or maternal gluconeogenesis, eventually helping the fetal brain. As a result, during gestation, a DF diet might help provide DHA for the fetal brain and MC-FAs for fetal adipose tissue storage, which will then be released later for ketogenesis, ALA sparing and DHA synthesis. Indeed, SC/MC-FAs are directly absorbed by the liver through the portal vein and then quickly β-oxidized, protecting n-3 PUFAs from oxidation and allowing them to be redirected toward conversion into the LC-PUFA n-3[57,73,74]. Moreover, SC/MC-FAs may more directly promote the synthesis of ω3 long-chain derivatives. Indeed, moderate myristic acid uptake (~1.2% of total energy) in humans increased plasmatic DHA[75]. This mechanism likely occurs through the enhancement of Δ6-desaturase activity, promoting conversion of ALA to DHA[76]. Dietary MC-FAs can also be transferred to milk. However, this is not the classical pathway of production because SC/MC-FAs are mostly synthesized de novo in the mammary gland from acetyl-CoA[77,78]. Indeed, an ancient interventional study demonstrated increased milk myristic acid (C14:0) in lactating women consuming high-fat dairy products or alpine butter[79,80]. Nonetheless, data on how maternal DF uptake might be reflected in the plasma or brain of offspring are limited.

Only one study showed that maternal DF intake in mice increased brain DHA accretion in offspring at 14 days postpartum[81].

After $D + 16$, infant mouse lemurs may begin to leave their nest box on their own[52]; however, this behavior is more common around the third week[46]. Diet diversification and solid food consumption begin at $D + 20$[46]. Between $D + 26$ and $D + 39$, the offspring attain nutritional and social independence and are considered weaned at $D + 35$ as suckling becomes rare[52]. Thus, we can expect that our animals would benefit directly from the DF diet when the Rotarod© test was performed at $D + 22$, especially for the visual discrimination test at $D + 30$. Hence, it was not possible for us to distinguish whether the enhanced long-term memory abilities at $D + 30$ originated from maternal or offspring diets. Both pathways of exposure to a DF diet may contribute to the observed phenotype. Indeed, on the one hand, maternal dairy fat intake during pregnancy and lactation preserves working memory during adulthood in mice, even though dairy fat is no longer consumed[29]. On the other hand, weaned young rats directly fed a dairy/vegetable fat mix containing 1.5% ALA and 14% LA displayed increased brain DHA accretion in comparison with those fed a vegetable fat diet containing 1.5% ALA and 16% LA[25,28]. Cognitive improvements at $D + 30$ might also arise from the earlier psychomotor development shown at $D + 8$. Indeed, earlier motor development allows the infant to experiment with the world sooner and enhances its ability to perceive and retrieve information about the environment and how it works[82]. This is supported by the fact that brain regions involved in early motor functions, such as the cerebellum and the prefrontal cortex, are also involved in cognitive control[83].

Obviously, strong physiological differences are noted between humans and mouse lemurs, and their nutritional needs are certainly not the same. In particular, human neonates are born with a considerable amount of subcutaneous fat stores, which are not observed in any other primates[84,85]. These fat stores are 3 to 4 times richer in DHA than those in adults and contain 8–10% SC/MC-FAs. Hence, they may continuously provide DHA and SC/MC-FAs for a certain amount of time even after weaning. Neonate mouse lemurs do not share this characteristic and may therefore benefit more from maternal dairy fat uptake. Moreover, numerous physiological data that confirm the actual exposure of neonates to DF, such as plasmatic and brain fatty acid analyses, are lacking. Nevertheless, our data suggest that replacing an important part of vegetable lipids with dairy lipids may be related to strong cognitive improvements and may constitute an appropriate vegetarian and lactose-intolerant diet. Indeed, butter does not contain lactose and is part of a lactovegetarian diet. On the other hand, the VF diet is closer to a vegan diet in terms of lipids, with 3.4% of lipids only derived from egg yolks. Our data suggest that a vegetarian diet including dairy products is more balanced and adequate to aid in psychomotor development than a diet incorporating most of its lipids as vegetable oils. DF provided mouse lemurs 21% of the total diet energy, which is equivalent to a butter intake of 61 g/day for men and 49 g/day for women based on daily energetic human needs[86]. This intake is 2 to 3 times higher than the average consumption of the French population (22 g/day), and thus may seem excessive[87]. Nevertheless, it is appropriate if this increase in intake is compensated for by a decrease in other commonly consumed fats, such as sunflower oil (linoleic acid) and red meat (long-chain saturated fatty acids such as palmitic and stearic acids), which are becoming increasingly common in the Western diet and are associated with adverse health outcomes when consumed in excess[88,89].

This study explored maternal dairy fat intake in association with the psychomotor and cognitive performance of offspring. The benefits for DF offspring were consistent across tasks, resulting in an increased number of successful neonates. Motor coordination, graviception and procedural memory were the cognitive domains most likely to be enhanced, whereas psychomotor and cognitive learning remained similar between the dietary groups. Among successful neonates, scores minimally differed according to diet. Finally, motor coordination scores were similar between DF and VF mouse lemurs at $D + 30$, demonstrating that DF neonates displayed earlier psychomotor development than VFs. The originality of this work relies on the use of a food approach, rather than a single-nutrient approach, which better reflects our diet and food behavior. This study suggests that a balanced

vegetarian diet including dairy lipids (i.e., a lactovegetarian diet) is more adequate for providing the nutrients needed for psychomotor development than a diet relying mostly on vegetable lipids. The effects of DF must first be validated by physiological biomarkers before proceeding with further research. Therefore, several issues might be addressed, such as 1) the phase of dietary intervention (pregestation, gestation, or lactation), which is more impactful for development; 2) the relative importance of maternal and infant diets in the effect observed; and 3) the possible life-long impacts when DF is sustained through aging.

## Methods
### Animals
Mouse lemurs were born, raised, and housed in the Brunoy colony of the French Museum National d'Histoire Naturelle (CNRS/MNHN in Brunoy, France; European Institutions Agreement no. E91–114.1). The experimental units were defined as both the animal and the litter from which it came. Because no previous study has investigated the impact of nutrition on the psychomotor development of mouse lemurs, the sample size could not be calculated a priori without leading to unreliable results. However, previous dietary interventions involving adult mouse lemurs with cognitive measures included between 6 and 12 animals per experimental group. Thus, because litter is the smallest experimental unit of the present study, a sample size of 11–12 litters was used. All experiments were performed in accordance with the Principles of Laboratory Animal Care (National Institutes of Health publication 86–23, revised 1985) and the European Communities Council Directive (86/609/EEC). The research was conducted under the approval of the Cuvier Ethical Committee (Committee number 68 of the 'Comité National de Réflexion Ethique sur l'Expérimentation Animale') under authorization number # 29784-2021021215198648 v3.

Multiparous female mouse lemurs (2 to 4 years old) were randomly assigned to one of the experimental groups using a computer-based random function and were fed the experimental diets (Supplementary Tables 5 and 6 for diet composition) 2 months before conception. They initiated their breeding season when the day length reached 14 h, which represents the long-day wet season in Madagascar. Light regime transitions from short-day to long-day conditions (and reciprocally) occur abruptly every six months since photoperiodic phenotype changes are more threshold-dependent and do not require gradual transitions to occur. The mouse lemurs were checked daily for oestrus starting 2 weeks after the photoperiod shift. If oestrus was confirmed, females were grouped with multiple males in the same cage for one week until a sperm plug was observed, which confirmed that breeding occurred, or until the closing of the vagina, which meant that the oestrus period had ended. Suspected fertilized females were then isolated in individual cages (50 × 60 × 70 cm³). Pregnancy was confirmed by palpation (~1 month). Gestation lasted between 57 and 65 days. The exact date of birth, sex, and litter size were recorded. The same protocol was repeated for 2 years until 50 pups were obtained, i.e., 25 per experimental group. A total of 22 females were selected for reproduction, 1 of which gave birth twice (representing 6 pups) over two years (thus one reproductive cycle per year). This female was fed an animal facility diet[90] during the interbreeding period (~6 months). Given that we cannot exclude the possibility that diet had long-lasting effects on this female, we included the mother identity in the statistical models (see the "Statistical analyses" section). The deaths of three infants were reported during the first three days of life: one in a DF litter (initial litter size = 4) and two in a VF litter (initial litter size = 2).

The animals were kept at constant temperature (22–25 °C) and relative humidity (55%) under a long-day photoperiod (14 h:10 h light/dark) with artificial white lighting (250 lux). Because the long-day period lasts six months, mothers and their offspring were under the same light regime throughout the entire length of the study. Cages were cleaned and densely enriched with new branches and foliage of *Prunus laurocerasus* every two weeks. Cages were equipped with at least two wooden nest boxes allowing the mother to temporarily isolate herself from her offspring. All mothers and their pups were checked and weighed on Days 1, 3, and 5 after birth followed by a weekly basis thereafter.

### Diet and feeding
Both experimental diets shared the same basis, i.e., a liquid constituting ~93% of the final mixture and made up of gingerbread, cereals, delipidated white cheese, eggs, and water (exact composition and recipe in Supplementary Table 5). The remaining 7% constituted the experimental lipidic input, providing as much as 92.6% of the total fatty acids. In the dairy fat-based (DF) diet, 75% of lipids were derived from butter, 16.7% from oleic acid-rich sunflower oil, and 8.3% from rapeseed oil. In the vegetable fat-based (VF) diet, 65% of lipids were derived from palm oil, 25% from oleic acid-rich sunflower oil, and 10% from rapeseed oil (Supplementary Table 6). Both diets were isocaloric and provided ~33% of the energy as lipids (Supplementary Table 5).

Animals were fed three times a week with 20 g of the final mixture described above per animal. Animals were given free access to water. On the first and second meals of the week, the mixture was accompanied by an apple slice (~10 g). On the third meal of the week, the mixture was accompanied by a banana slice (~10 g) and a cucumber slice (~10 g). During the second month of gestation, mothers were given an extra 10 g of the diet mixture. During lactation and until weaning, mothers were given 20 g of feed plus 10 g/pup.

Six samples from each DF and VF mixture were analyzed for fatty acid composition in Dr. Plourde's laboratory at the Centre de Recherche sur le Vieillissement (Sherbrooke, Qc, Canada). For each sample, 300 µL of a standard solution (0.016 g of triheptadecanoin—C17:0, triglyceride form—in 10 mL of chloroform) was added to 25 mg of the diet mixture. Lipids were extracted according to the method of Folch et al.[91] with 10 mL of chloroform/MeOH (2:1), after which the vials were vortexed and left for one hour in the dark. After centrifugation at 1500 rpm for 10 min, the bottom phase (chloroform + lipids) was extracted, dried under nitrogen, saponified with 100 µL of NaOH (0.5 M) in MeOH, and methylated with 0.5 mL of BF₃ solution (12–14% in methanol) as a biocatalyst[92]. Fatty acid methyl esters were then analyzed using gas chromatography coupled to a flame ionization detector as described by Chevalier et al.[92].

As expected, the DF diet mainly featured a greater concentration of all SC/MC-FAs compared with the VF diet (Mann-Whitney test, W = 36, $p = 0.002$) (Supplementary Table 7). In particular, the VF diet completely lacks fatty acids with shorter chains, i.e., caproic (C6:0), caprylic (C8:0), and capric (C10:0) acids, whereas myristic acid (C14:0) is tenfold less concentrated in the VF diet (Mann-Whitney test, W = 36, i = 0.002). In contrast, the DF diet was depleted of saturated long-chain palmitic (C16:0) and arachidic (C20:0) acids compared with the VF diet (Mann-Whitney tests, W = 0, $p = 0.002$ and W = 0, $p = 0.003$, respectively). The VF diet is richer in monounsaturated fatty acids, mostly due to its higher concentration of oleic acid (C18:1 n-9) (Mann-Whitney test, W = 0, $p = 0.002$). The second main feature of the DF diet was its lower LA/ALA ratio (Mann-Whitney test, W = 0, $p = 0.002$). However, both the DF and VF diets are completely depleted of long-chain PUFAs. DF also displayed interesting features in minor fatty acids, such as its richness in odd-chain saturated and unsaturated fatty acids (C15:0, C15:1, C17:1) (Mann-Whitney tests, W = 36, $p = 0.003$; W = 33, $p = 0.010$ and W = 30, $p = 0.028$, respectively) and trans fatty acids (C18:1 t n-7, C18:1 t n-9) (Mann-Whitney tests, W = 0, $p = 0.002$, and W = 30, $p = 0.028$, respectively).

**Psychomotor tasks.** Four psychomotor and cognitive tasks were designed, some of which were adapted from Fox's tasks[93] and used as a standard to investigate the ontogeny of reflexes and behavior in mice. We hypothesized that these tasks would also work on mouse lemurs, whose morphology resembles that of mice, after taking their specific arboreal ecology into account. The following major constraints were considered in the design of tasks specifically dedicated to mouse lemur pups: (1) all tasks must be quick (<1 h) to avoid fatigue and too long of a separation from the mothers, but tasks must be sufficiently difficult to demonstrate variability between animals; (2) psychomotor tasks should require a limited preliminary learning phase; and (3) most of the tasks should use naturally expressed abilities (such as arboreality). All devices were

https://doi.org/10.1038/s42003-024-06255-w **Article**

washed with white vinegar between trials. When 8-day-old animals were handled, a heating lamp was used to prevent a decrease in body temperature. All tasks were performed between the 11th and 14th hours of the day, i.e., during the three-hour period preceding light extinction. This choice constituted a compromise between the following: (1) the need for investigators to work in the middle of the night for three years was unrealistic; (2) despite their nocturnal ecology, mouse lemurs can still perform cognitive tasks during this period because they begin to anticipate the incoming night by increasing their activity[94]. On the same day, the order of the animals within a litter (in the case of twins or triplets) and between litters were randomized. All tasks were performed in an adjacent room to the animal facility, while the mother remained in the cage. During a testing session, the animal's nest box was used as a positive reinforcement for the tested animal, while its littermates (if any) remained in a neutral nest box until their turn. Finally, the investigator was not blinded to the diet of the animals because it was the same person who fed the animals and took care of them for at least one year of this 2-year study until additional staff were recruited.

**Negative geotaxis.** Eight-day-old animals (D + 8) were placed with their noses on the center of a white 60°-inclined polyvinylchloride (PVC) surface (25×30 cm²), with the head pointing downward (Supplementary Fig. 1, insert A). This task used the natural behavior of neonates to flip over when positioned this way, preventing injuries that may occur during a fall. We measured the time required for the animal to reach the horizontal (Th) and vertical (Tv) marks. The experiment was repeated thrice, and the best Th and Tv were recorded. Between each trial, the animals were kept in the investigator's hands for a 1-min resting phase. The maximum trial duration was fixed at 3 min. Trial outcomes were also recorded as "fall" (if the animals fell before reaching the marks), "unreached" (if the animals failed to reach the marks before the 3-min period), and "success" (if the animals reached the marks within the time frame). The proportion of successful trials was recorded for each animal.

**Horizontal rod: early psychomotricity.** After testing for negative geotaxis, D + 8 animals were placed on one end of a wooden rod (length=14 cm, ø=0.5 cm) and suspended 17 cm above the ground with white PVC boards (Supplementary Fig. 1, insert B). The animal's own nest box was placed at the other end of the rod, which was thought to motivate the animal to cross the rod and reach its nest. The procedure consisted of 4 steps. The first procedure was a habituation procedure. The animal was placed on the rod immediately in front of the nest box, which consistently ended with the animal stepping in its nest box within less than 1 min. The following steps consisted in three trials with increasing levels of difficulty: the animal was placed on the rod with the nose 11 cm, 22 cm and 33 cm, separately, from its nest box. For each trial, if the animal did not move for a 30-s period, we tried to stimulate it by slightly jiggling the rod. The time to reach the nest box was recorded for each level of difficulty. Reaching the nest box was defined as the complete entrance of the head inside the nest box. The animal was kept inside its nest box for a 1-min resting phase. The maximum trial duration was fixed at 4 min. Trial outcomes were also recorded as "success" (if the animals reached the nest box within the time frame) or "fail", which was categorized as "fall" (if the animals fell before reaching the nest box) or "unreached" (if the animals failed to reach the nest box before the 4-min period). Jumping or walking backward were counted as "other" because these outcomes occurred infrequently and thus could not be analyzed as such. Baby mouse lemurs displayed two extreme locomotor strategies: they walked straight to the nest box with relatively good motor coordination or they cyclically rolled over while hanging on to the rod with their back facing the ground and climbed back on the rod afterward, which made them progress slowly to the nest box. The latter moves were called "helicoid movements" and were recorded as well.

**Rotating rod: late psychomotricity (day + 15, + 22 and + 30).** The device used to assess D + 15, D + 22 and D + 30 psychomotricity, collectively referred to as "late psychomotricity", consisted of an accelerating motor-driven rotating rod (Rotarod©, Model 7750 Ugo Basile, Italy, ø=5 cm) (Supplementary Fig. 1, insert C). In the preliminary habituation phase, which occurred exclusively at D + 15, the animals were placed on the rod at a constant speed of 5 rotations per minute (rpm) for 1 min. Then, the testing phase consisted of a maximum of 5 10-min trials, each of which was divided into a 5-min constant acceleration phase from 10 rpm to 30 rpm followed by a 30 rpm constant speed phase for 5 min. An unfolded hemp towel was placed under the rod to gently receive the animal in case of a fall. The trial stopped when the animal fell, jumped, or remained hanging on the rod for 3 rotations without moving. If the animal did not stay on the rod for more than 3 s, the trial was repeated up to 6 times, after which the trial was counted as "refusal". If an animal refused to participate in one of the 5 trials, it was excluded. This occurred only at D + 30 (n = 1 DF and n = 2 VFs). We measured the time spent by the animal on the rod for each trial and recorded only the maximum time among all trials. Trial outcomes were also recorded as "fall", "jump", "hung on for 3 rotations", "refusal", or "maximum" (if the animal stayed on the rod for 10 min).

**Visual discrimination task in the open field test (day + 15 and + 30).** After being subjected to the Rotarod© test, D + 15 and D + 30 animals were subjected to a visual discrimination task. The visual discrimination device consisted of a white PVC box (40 × 40 × 34 cm³) with a laterally interchangeable face drilled with either 1 or 4 square-shaped exits, all of which were regularly spaced 4.5 cm away from the adjacent exits (Supplementary Fig. 1, insert D). The leftmost exit was covered by a light black cloth that could be easily pushed by the animals and topped by a green circle-shaped cue. The individual's own nest box was placed behind this exit so that the animal could reach it by going through the cloth. The other (fake) exits were covered by the same cloth as well. However, the cloth was strongly tapped, preventing the animal from going through. The position of the correct exit never changed. The experiment consisted of 2 phases:

Habituation phase: The lateral face only possessed one exit. The animal had to learn that it could reach the nest box by going through the tape. To do so, three trials were performed. For each trial, we placed the animal at three successive positions. The nose was positioned 5 cm, 15 cm and 28.3 cm from the leftmost exit, and all three points were aligned on a diagonal of the basis face (the last one being the exact center of the box). For each trial, the animal had 30 sec to reach its nest box; if it did not, the investigator helped him understand by pushing him to the nest box and forcing him to go through the cloth. Each trial was followed by a 1-min resting phase in the nest box.

Testing phase: The lateral face contained all 4 exits. The animal had to remember the position of the correct exit despite the presence of interfering signals from the 3 other exits. The testing phase consisted of 3 trials at a maximum of 5 min each. The animal was positioned at the center of the device, as in the third trial of the habituation phase. The animal was kept inside its nest box for a 1-min resting phase following each trial. For each trial, if the animal did not move for a 30-s period, we tried to stimulate it by slightly knocking the bottom of the device with our fingers. The time to reach the nest box was recorded for each trial. Reaching the nest box was defined as the complete entrance of the head inside the nest box. Animals that did not reach the nest box within the time frame were assigned a maximum time of 300 sec.

**Video analysis**

All trials were videotaped with a GoPro HERO4 (GoPro, Inc. CA, USA). For each task, the timer started once the animal was freed from the hands of the investigator. Videos were analyzed with Noldus EthoVision XT software, which allowed us to obtain precise durations (1/10th of a second). One video from the negative geotaxis task and eight videos from the horizontal rod task

were inoperable because of investigator misconduct regarding the protocol and were therefore excluded from the analysis.

## Statistics and reproducibility

All statistical analyses were conducted using R (version 4.2.2) software. Our data often displayed repeated measures on the same animal, overdispersion and nonnormality of residuals. To consider these features, we analyzed our data with generalized linear mixed models (GLMMs) using the *glmmTMB* package. The identities of each animal and of each mother were used as random factors. Gamma distributions and a log link function were used for temporal measures (e.g., time to reach the nest box). Beta regressions and a logit link function were used for overdispersed binary or proportion measures (e.g., %failures versus %success). Negative binomial distributions and a log link function were used for overdispersed count measures (e.g., number of exits visited). For some variables, our data displayed an excess frequency of extremum values of the distribution (e.g., maximum or minimum time reached). If the variable $X$ displayed an excess of 0 (excess of the maximum value referred to as *max*), we modeled $X$ ($X$ - *max*) as a zero-inflated distribution, which allowed us to separately analyze the distribution of $X$ without extreme values and the number of extreme values for itself. Models were systematically confirmed by checking for overdispersion and deviation and normality of residuals using the *DHARMa* package. When heteroscedasticity between groups was identified, the model was corrected using the *dispformula* parameter. Diet (DF or VF) was the main explanatory variable, but we also controlled for other confounding factors, such as sex, body weight and litter size. The best-fit models were selected using the corrected Akaike information criterion for small samples (AICc). Pairwise comparisons between modalities of the same variable were performed using the *emmeans* package, and $p$ values were adjusted with the multivariate $t$ method. Specifically, for dietary fatty acid analysis, Mann-Whitney tests were used because the measurements were performed on independent samples. Reporting of GLMMs results includes the following: the z-value statistic, the back-transformed estimated coefficient $\hat{\beta}$ (or odds ratio OR when a binomial distribution or beta-regression was used) with its 95% confidence interval between brackets, and the $p$ value rounded to the nearest third digit after the decimal point. Differences were considered significant when $p < 0.05$. Animals that refused to complete a task were excluded from the analyses. Descriptive statistics often depict the mean ± standard error, except for Supplementary Tables 1 and 2, for which the median ± inter-quartile range was used because the sample size was reduced and extreme values would have had too strong of an influence.

## Reporting summary

Further information on research design is available in the Nature Portfolio Reporting Summary linked to this article.

## Data availability

The data that support the findings of this study are available in the Supplementary Data file.

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

## Acknowledgements
We sincerely thank Martine Perret and Aude Anzeraey for their assistance in animal reproduction and selection and their valuable advice. We thank Isabelle Hiron, Lauriane Dezaire and Sandrine Gondor for caring for and feeding the animals. We finally thank Valentin Célérier, Eugénie Mortessagne, Anaïs Pichevin, Lucas Fructus, Crystal Morin, Cécile Jartoux and Florian Medetognon Benissan for their support during the experiments. We finally thank Amandine Ligneul and Marion Lemaire for providing helpful insights after reading this article.

## Author contributions
Y.C. contributed to the investigation, data collection, formal analysis, project administration, and writing—original draft and revisions; C.B., C.M., B.D., M.T., and F.P. contributed to conceptualization, methodology, and writing—revisions; A.V. contributed to diet fatty acid analysis and writing—revisions; M.P. contributed to writing—revisions.

## Competing interests
Yohann Chaudron, Constance Boyer, Corinne Marmonier, Bernadette Delplanque, Mohammed Taouis and Fabien Pifferi declare the following competing interest: this study was financially supported by the Centre National Interprofessionnel de l'Economie Laitière (CNIEL). Annick Vachon and Mélanie Plourde declare no competing interests.
