## [Peer Review File · Communications Biology]

Reviewers' comments:

Reviewer #1 (Remarks to the Author):

In this article, Chaudron et al. used a grey mouse lemur model to study effects of maternal dietary exposure to dairy fat during gestation and lactation on the performance of the offspring in a behavior test battery.

Overall, the work is novel and interesting and the results represent a potential application in dairy industry. Nevertheless, I would like to raise some points, which I feel should be considered in the manuscript.

Introduction:

1. I suggest authors to improve the introduction with more information regarding maternal/ infant nutrition and motor development as the majority of tasks in the used test battery was depends on motor function.

2. Authors raise two main hypotheses that could explain the lack of evidence for associations between maternal DHA supplementation and infant cognitive outcomes in humans, one of which being that in none of the studies supplementation was initiated before 20 weeks of gestation. While the idea that effects of DHA supplementation could be restricted to a critical phase during development it is an interesting point by itself, it is not fully clear why this particular point was raised in relation to the current work. The study design does seem to contribute to studying this hypothesis, nor does the point come back in the discussion section. Moreover, the reference used does not appear support the specific hypothesis. If the authors want to discuss this point in the manuscript, I suggest to at least include some more references from (preclinical) studies to support the hypothesis and to further build the argument.

Results:

3. Diet fatty acid composition: I praise the authors for testing and disclosing the fatty acid composition in the experimental diets, especially since several batches per diet appear to have been used during the study causing some variability. It is somewhat confusing however that the composition of the two diets is statistically compared and results of that are presented in the results section. The dietary fatty acid composition is not an outcome measure of the study unless one of the research correct? I suggest moving this entire part to the methods.

4. Offspring characteristics: In the methods it is specified that bodyweights were also collected at 1, 3 and 5 days after birth. were there any effects of diet type on bodyweight at these timepoints? Same question applies to birth date (gestation length?) sex and litter size. The matching of information provided in methods and results section, and more complete reporting supports transparency. Please add a (supplementary) table to results or methods section to disclose exactly how many litters were present per diet group and in what litter compositions (sex & number of pups).

Discussion:

5. In contrast to rats and mice, the grey mouse lemur is not a very common species to use for nutritional intervention studies. It can be expected that many readers will therefore lack generic knowledge about e.g. the developmental trajectory and physiology of this species that may be needed to place the results of the current study in context. I suggest the authors to add more species-specific information in relation to the discussion (or elsewhere when not relevant for discussion) to help understand the study.

6. For example, diets were provided to the mothers during gestation and lactation. How many days is the normal lactation period of the grey mouse lemur? Are the offspring exclusively fed on mothers milk up to the 30 days study period or do they also get some exposure to the experimental diet themselves? The potential underlying mechanisms may be different for maternal interventions and infant interventions. Obviously, the applications of the findings will also be quite different when translated to maternal interventions or infant intervention.

7. Along with this critical discussion point some more background information should be provided on what is known about the transfer of the specific fatty acids of interest / that are different between diets (LA, DHA and in particular the SC/MC-FAs) across the placenta and into mothers milk. The underlying mechanisms by which the effects of diet are mediated appears to be higher DHA levels in the mother. Other than DHA, do the authors propose that the dietary fatty acid composition of the mother is directly translated into fatty acid composition available to the offspring and if so, would direct exposure of offspring to the other fatty acid levels in the dairy diet cause similar effects?

8. The differences between diet groups in offspring bodyweight towards the end of the study deserves some more discussion. What would be considered a normal/healthy growth rate for the mouse lemur and is the lower bodyweight observed in the DF vs VF group considered beneficial and why? It would be informative to add a few lines to the discussion describing what is known in literature about some of the specific dietary fatty acid species mentioned and effects on early life growth rate and or obesity risk. What is known about the metabolic changes in this animal species in response to changes in dietary lipids? The authors address that bodyweight correlates with strength, which is presumably based on muscle mass. Could it however also be that the difference in bodyweight observed between groups is due to differences in fat mass? LA is known to be adipogenic, animals that have higher fat mass relative to total body mass may in fact have weaker muscle strength and will fall more easily? Moreover, authors suggest that differences in litter size between groups may underlie differences in bodyweight observed. What is the normal/typical litter size of the grey mouse lemur? Could a litter of 1 pup be seen as unintentional postnatal overfeeding model as is used in rodents as obesity model?

9. Authors conclude that in particular motor function is improved. Missing in the discussion are a few lines on what is known in literature on effects of dairy and/or the specific fatty acids on motor function (see also the comment to the intro)?

10. Is the level of maternal behavior/care towards pups early in life likely to influence motor and cognitive development / function of the grey mouse lemur similar to other species? If so, were these

things monitored in the current study and are there reasons to assume that their maternal diet would have affected this? Same could be said about early social experience (e.g. play, competition) in the presence or lack of littermates. If indeed applicable to this species I suggest the authors to briefly acknowledge that next to diet, other environmental factors could have contributed to the results observed.

11. In the last paragraph of the discussion, the authors mention for the first time vegetarian / lactose free diets. Suggest to strongly emphasize the relevance of the current work in relation to the increasing popularity of plant based diets (both maternal and infant nutrition). Possibly already raise this point in the introduction.

Conclusions and perspectives:

12. In line with the comment to the introduction, the experimental design used in the current study does not support the statement that the strong modifications might originate from the nutritional protocol starting 2 months prior to conception. Results could also have been mediated by the intervention during lactation. I suggest to either remove this statement or provide a better rationale why the current study design, possible with more background knowledge provided, is suitable to support this statement.

13. Next to physiological biomarkers, suggestions for future studies would then be to study which phase (pre-gestation or lactation) would be more important.

14. It remains somewhat unclear if DF is presented as nutritional solution for mothers or for infants? In case presented for infants it would mean that one critical step here would be to study intervention in infant, not via mother.

Methods:

15. In general, I recommend the authors to carefully review the ARRIVE guidelines at arriveguidelines.org/arrive-guidelines and making a few simple additions to the methods & results section accordingly (see the examples in Essential 10) to improve overall reporting style. Particular points of attention include amongst others the definition of experimental unit (see also comment below), randomization/blinding and sample size justification.

16. In addition, as many readers will not have any experience with the grey mouse lemur as model organism, a somewhat more extensive description of relevant species specific information in relation to the housing, husbandry and procedures would be appreciated (see also some of the comments below).

Reproduction:

17. The light regimes used during the study is unclear. Was there a gradual build up in length of light period to reach 14 hours and start breeding cycle? Were pregnant and lactating dams and their offspring also kept on the 14 hours light regime.

18. In case females were used for multiple breeding rounds: How many breeding cycles/year were initiated and what diets were provided in between breeding/ intervention periods?

Psychomotor tasks:

19. Is the grey mouse lemur a diurnal/ nocturnal species and what in what phase did behavior testing take place? Can you describe the test order on a testing day when multiple tests were performed? Was test order of individuals randomized and was there habituation to the room in which testing took place?

20. Some tests use the nest box as motivation to escape to. Was this a neutral nest box or was it the individual's own nest (scent might be important). What happened to the mother and littermates during testing of individuals?

Statistical analysis:

21. Please explain how sample size was determined.

22. The mothers were subjected to the diet interventions during pregnancy and lactation, yet individual pups were used as the experimental units rather than mother/litter. Next to variation between dietgroups, It can be expected that there is some variation between mothers within diet groups as not all mothers/litters are exactly the same. For example there was variability in littersize as described in the results and littersize may by itself affect outcomes through various mechanisms as mentioned above. This calls for inclusion of litter of origin as random factor to the statistical models used. Same applies to the (diet) batch as there were multiple batches used. Please confirm that these factors were included or re-run the analysis with inclusion of these critical factors.

Figures

23. Figure 5 A: It is impossible to count the dots that represent the number of animals per group succeeding and not succeeding in this figure as some of the dots are overlapping. Consider adapting the layout to allow people to discriminate individual dots. Alternatively, add the actual numbers in the figure.

24. Figure 7 and 8: see previous comment

Reviewer #2 (Remarks to the Author):

This paper seeks to demonstrate dairy fat (DF) intake in mothers, pre- and post-birth while nursing, provides for improved psychomotor and cognitive performance in infant mouse lemurs than supplementing with vegetable-based fats (VF). Based on the data presented, the infant mouse lemurs

did show greater neuromotor control and cognitive development based on visual discrimination tasks. While I was enthusiastic about this concept, I did not see a clear significance of this work presented. What is the reason for this work? It appears that the authors want to demonstrate the need for women to continue to use dairy products during pregnancy and nursing instead of relying on vegetable fats during this time. Although the abstract says: "Our results support further research on the impacts of dairy fat on cognitive performance after complete cessation of breastfeeding." This was not presented in the manuscript. Throughout the Abstract, Introduction, and Discussion, I looked for a clear impact of this work, but did not find it.

One of my main concerns is in the experimental design. The study examines maternal supplement with either DF or VF but there is no control condition. What is the neural and cognitive performance of infants without maternal supplement? How much did the DF or the VF improve normal neural and cognitive functioning? While the DF contained A and D vitamins, did the VF supplement? The studies relied upon 6 females during multiple pregnancies of litters (1,2, or 3 in a litter) for the subjects to test. Does this limit the power of the results? Many of the results exhibited large error bars making differences look less effective. What was the effect size for the analyses?

This study could have been more impactful if it was written with a clear reason. Are the authors concerned that women will not use DF? Is there a concern that mothers or infants will only receive VF as in a vegan diet?

We would like to thank the reviewers and the editor for their encouraging and fruitful comments. We took them into account, and believe they really helped improve the manuscript. In particular, we significantly improved and clarified the rationale behind our study, and the needs for these results. Please see below our answers to each comment, and let us know if the corrections that have been made match your expectations, or if you have any further question or comment. All changes in the manuscript are shown in blue.

Concerning the comments made by reviewer 1:

- ***Introduction:***

1. I suggest authors to improve the introduction with more information regarding maternal/ infant nutrition and motor development as the majority of tasks in the used test battery was depends on motor function.

Lines 58-70: We have added a statement about motor development and perinatal DHA supplementation. We sincerely thank the reviewer for this suggestion since the literature tends to demonstrate supplementation effects only for early motor performances, but not for the late ones, which is consistent with our findings.

2. Authors raise two main hypotheses that could explain the lack of evidence for associations between maternal DHA supplementation and infant cognitive outcomes in humans, one of which being that in none of the studies supplementation was initiated before 20 weeks of gestation. While the idea that effects of DHA supplementation could be restricted to a critical phase during development it is an interesting point by itself, it is not fully clear why this particular point was raised in relation to the current work. The study design does seem to contribute to studying this hypothesis, nor does the point come back in the discussion section. Moreover, the reference used does not appear support the specific hypothesis. If the authors want to discuss this point in the manuscript, I suggest to at least include some more references from (preclinical) studies to support the hypothesis and to further build the argument.

The argument has been accordingly removed from the introduction (and have been replaced by the argument about differential effect on motor development depending on the timing of testing).

- ***Results:***

3. Diet fatty acid composition: I praise the authors for testing and disclosing the fatty acid composition in the experimental diets, especially since several batches per diet appear to have been used during the study causing some variability. It is somewhat confusing however that the composition of the two diets is statistically compared and results of that are presented in the results section. The dietary fatty acid composition is not an outcome measure of the study unless one of the research correct? I suggest moving this entire part to the methods.

As suggested, we have moved the diet fatty acid composition part to the methods. We have kept the statistical analysis to demonstrate that the fatty acid composition did not differ between the different batches, but we can remove it if needed.

4. Offspring characteristics: In the methods it is specified that bodyweights were also collected at 1, 3 and 5 days after birth. were there any effects of diet type on bodyweight at these timepoints? Same question applies to birth date (gestation length?) sex and litter size. The matching of information provided in methods and results section, and more complete reporting supports transparency. Please

add a (supplementary) table to results or methods section to disclose exactly how many litters were present per diet group and in what litter compositions (sex & number of pups).

We added a **supplementary table (Table B)** with gestation length, sex ratio, litter size and number of litters according to the diet. None of these demographical variables differ significantly according to diet. While at first sight the sex ratio may seem to vary a lot, statistical tests do not support that the sex ratio is biased. This may be the consequence of our little sample size. However, we systematically took sex into account in our statistical models, and never saw an effect of sex.

Lines 127-131: we call for this Table C in the “Results – Offspring characteristics” section.

We also added a **supplementary table (Table A)** with the body mass at D+1, D+3 and D+5. However, these data have not been recorded for all animals (see sample size) since: 1) this was not a primary outcome measure of our study and 2) the dates happen to fall on the week-end, with limited staff on the site and limited interventions towards the animals. Thus, for transparency we propose this table but we doubt of its scientific interest. Nonetheless, we did analyze statistically these results. With all the precautions taken due to sample size variation, we did not see a significant difference in body mass at these ages, although the median was higher in VF group at D+3 and D+5 (non-significant). This is consistent with the absence of difference at D+8 and D+15 (Figure 2), the differences beginning to be significant only from D+22.

- **Discussion:**

5. In contrast to rats and mice, the grey mouse lemur is not a very common species to use for nutritional intervention studies. It can be expected that many readers will therefore lack generic knowledge about *e.g.* the **developmental trajectory** and **physiology** of this species that may be needed to place the results of the current study in context. I suggest the authors to add more **species-specific information** in relation to the discussion (or elsewhere when not relevant for discussion) to help understand the study.

Line 95: we added “nocturnal” in the introduction;

Lines 257-260: referential for acquisition of crawling and climbing help argument the delay psychomotor development of VF animals

Lines 270-274: referential for 3D vision acquisition and use of olfactory cues

Lines 348-353: we added information related to lactation and weaning timing, and deduced that, at least for the D+8 and D+15 tasks, performances depended only on maternal diet and not the offspring diet.

Besides this added information, we wanted to bring to attention that information about psychomotor development is rare in mouse lemurs, excepted one study about the ontogeny of locomotion (Bourlinguez-Ambroise, 2019) and have been cited three times to set a locomotor referential. We are afraid there is no more study about mouse lemur psychomotor development during the first 30 days of life. Thus, we believe our study brings a helpful referential regarding psychomotor and cognitive development for upcoming studies.

6. For example, diets were provided to the mothers during gestation and lactation. How many days is the normal lactation period of the grey mouse lemur? Are the offspring exclusively fed on mothers milk up to the 30 days study period or do they also get some exposure to the experimental diet themselves? The potential underlying mechanisms may be different for maternal interventions and

infant interventions. Obviously, the applications of the findings will also be quite different when translated to maternal interventions or infant intervention.

Lines 349-353: we added information related to lactation and weaning timing, and deduced that, at least for the D+8 and D+15 tasks, performances depended only on maternal diet and not the offspring diet.

Along with the previous added discussion, we added lines **398-416** a discussion related to either maternal contribution (before weaning) or juvenile contribution (after weaning) might explain the differences observed at D+22 and D+30

7. Along with this critical discussion point some more background information should be provided on what is known about the transfer of the specific fatty acids of interest / that are different between diets (LA, DHA and in particular the SC/MC-FAs) across the placenta and into mothers milk. The underlying mechanisms by which the effects of diet are mediated appears to be higher DHA levels in the mother. Other than DHA, do the authors propose that the dietary fatty acid composition of the mother is directly translated into fatty acid composition available to the offspring and if so, would direct exposure of offspring to the other fatty acid levels in the dairy diet cause similar effects?

Lines 365-376: We added supplementary information regarding the placental transfer of ALA and DHA

Lines 377-384: We added supplementary information regarding the placental transfer of MC-FAs and ketones

Lines 398-416: We discuss the effects of exposure through maternal diet (indirectly) or offspring diet (directly)

8. The differences between diet groups in offspring bodyweight towards the end of the study deserves some more discussion. What would be considered a normal/healthy growth rate for the mouse lemur and is the lower bodyweight observed in the DF vs VF group considered beneficial and why? It would be informative to add a few lines to the discussion describing what is known in literature about some of the specific dietary fatty acid species mentioned and effects on early life growth rate and or obesity risk. What is known about the metabolic changes in this animal species in response to changes in dietary lipids? The authors address that bodyweight correlates with strength, which is presumably based on muscle mass. Could it however also be that the difference in bodyweight observed between groups is due to differences in fat mass? LA is known to be adipogenic, animals that have higher fat mass relative to total body mass may in fact have weaker muscle strength and will fall more easily? Moreover, authors suggest that differences in litter size between groups may underlie differences in bodyweight observed. What is the normal/typical litter size of the grey mouse lemur? Could a litter of 1 pup be seen as unintentional postnatal overfeeding model as is used in rodents as obesity model?

Lines 296-335: we added a discussion about the offspring weight, regarding multiple factors (maternal investment, litter size, sex ratio, anti-adipogenic effect of SC/MC-FAs and ALA + the adipogenic effect of LA). In this regard, we added a referential of demographic characteristics of mouse lemur development. As you will see, body weight at D+30 is slightly higher than expected, and is associated with higher number of single neonates, probably due to 1) the young age of mothers used in this study and 2) death of one neonate in two twin litters. Thus, we added more information in the Methods section about litter size and early mortality (**lines 497-499**), precisising that litter size at birth did not differ between dietary groups but finally differed due to mortality in the twin litters. A litter of 1 pup is not unusual, actually representing almost 10% of the litters in our colony, and is even more frequent in young mothers. In this regard, in our opinion, a litter of 1 pup can not be seen an unintentional postnatal overfeeding model.

Very few dietary lipid interventions have been led on mouse lemurs, and only in adults or aged animals, never perinatally. In adults and aged animals, n-3 LC-PUFA intake (EPA + DHA) did not modify body mass. Thus, this study is the very first to bring to light neonate's body weight changes in relation to diet, which are consistent with what have been found in rodents. However, we did not measure fat and lean mass, as suggested by Reviewer 1, which we agree would have been helpful in interpreting the motor differences between dietary groups. However, it is worth noticing that psychomotor performance already differed at D+8 and D+15, at the time body mass did not differ between dietary groups (which does not imply that fat mass and lean mass did not differ either).

9. Authors conclude that in particular motor function is improved. Missing in the discussion are a few lines on what is known in literature on effects of dairy and/or the specific fatty acids on motor function(see also the comment to the intro)?

Lines 287-294: Since introduction has been already improved with references and that there is not a lot of studies about motor development and n-3 PUFA intake, we only added a statement in the discussion about the fact that our results are consistent with the differences seen for early motor development milestones but not the late ones

10. Is the level of maternal behavior/care towards pups early in life likely to influence motor and cognitive development / function of the grey mouse lemur similar to other species? If so, were these things monitored in the current study and are there reasons to assume that their maternal diet would have affected this? Same could be said about early social experience (e.g. play, competition) in the presence or lack of littermates. If indeed applicable to this species I suggest the authors to briefly acknowledge that next to diet, other environmental factors could have contributed to the results observed.

Lines 296-346: along with the discussion about the body weight, we brought a discussion about maternal investment (296-330) and social environment (335-346) as suggested. Although very few information is available regarding early social experience (and no data about the interaction between early social environment and psychomotor development), we obviously acknowledge the potential impacts of these factors.

11. In the last paragraph of the discussion, the authors mention for the first time vegetarian / lactose free diets. Suggest to stronger emphasize the relevance of the current work in relation to the **increasing popularity of plant based diets** (both maternal and infant nutrition). Possibly already raise this point in the introduction.

Lines 40-55: we improved the introduction with the consequences of increased intake of vegetable oils during the last decades

Lines 83-92: we improved the rationale about the reasons we led this study

Lines 426-433 and 452-454: we specified the differences between DF and VF diets in relation to human food typology, with the DF depicting a lactovegetarian diet and the VF diet depicting a vegan diet regarding lipids

Conclusions and perspectives:

12. In line with the comment to the introduction, the experimental design used in the current study does not support the statement that the strong modifications might originate from the nutritional

protocol starting 2 months prior to conception. Results could also have been mediated by the intervention during lactation. I suggest to either remove this statement or provide a better rationale why the current study design, possible with more background knowledge provided, is suitable to support this statement.

We agree with this remark and removed the statement from the conclusion

13. Next to physiological biomarkers, suggestions for future studies would then be to study which phase (pre-gestation or lactation) would be more important.

We agree and incorporated this remark in the perspectives (**lines 456-457**).

14. It remains somewhat unclear if DF is presented as nutritional solution for mothers or for infants? In case presented for infants it would mean that one critical step here would be to study intervention in infant, not via mother.

The study is a nutritional solution for infant development, but through the mother since a large part of the brain development occurs during gestation and lactation. Our aim was to simulate a situation as natural as possible, without interruption of breastfeeding and infant formula supplementations. As it is discussed lines 349-353, at least the psychomotor performances at D+8 and D+15 do not depend on infant direct nutrition since infants depend only on maternal milk until the third week. It would not be possible to study the effects of DF on these early psychomotor performances if the diet was directly given to infants, since it would mean mother-infant segregation before weaning, which 1) is difficult to accept ethically in Primates, and 2) might be deleterious for infants. However, we acknowledge that in a future study, testing DF intake only after weaning would be an interesting perspective to delineate the effects of maternal vs juvenile diets. Thus, we added this statement in the conclusion (**line 458**)

- **Methods:**

15. In general, I recommend the authors to carefully review the ARRIVE guidelines at arriveguidelines.org/arrive-guidelines and making a few simple additions to the methods & results section accordingly (see the examples in Essential 10) to improve overall reporting style. Particular points of attention include amongst others the definition of experimental unit (see also comment below), randomization/blinding and sample size justification.

We thank you for the valuable attached guidelines, and tried to comply with them when possible:

Lines 465-466: We specified that both litter and individual identity were used as experimental units in the “Method – Animals” section. Indeed, 1) the diet was provided to the mother, which provided it to the offspring through the umbilical cord, and through milk, making the littermates belonging to the same experimental unit; 2) the younglings began to feed on the experimental diet by themselves around the third week, making each individual an experimental unit as well since cognitive outcomes continued to be recorded at D+22 and D+30

Lines 479-480 and lines 571-573: We specified that multiparous females were randomly assigned to each experimental diet and that daily testing order of each neonate within a litter was randomized as well.

Lines 576-578: We specified that the investigator was not blinded when conducted the experiments in the “Methods – Psychomotor tasks” section. Indeed, it was the same investigator that made the diets,

fed the animals, take care of them and tested them for psychomotor outcomes, at least for a large part of the study, until we were able to recruit animal care staff one year after the beginning.

We regret that sample size could not have been *a priori* rigorously calculated, since no previous data exist regarding the effect of a diet on psychomotor performances during early development. We do have data during adulthood, however 1) we expected the performances and the variability to differ a lot in neonates; 2) the apparatuses were not adapted to neonates. Then, any attempt to calculate an *a priori* sample size for our study would lead to unreliable results. However, previously dietary interventions led in our lab, which evaluated cognitive outcomes as well, enrolled usually between 6 and 12 animals per experimental condition (Dal-Pan et al., 2011; Languille et al., 2012; Marchal et al., 2012; Pifferi et al., 2015, 2018; Royo et al., 2018; Vinot et al., 2011). Thus, since our experimental unit is not only the animal but also the litter, we planned to enroll around 12 litters per experimental condition in our study (actually n=11 litters in the DF group and n=12 litters in the VF group), which led to 25 animals per group. For the sake of transparency, this justification has been added in the “Methods – Animals” section **(line 466-471)**

16. In addition, as many readers will not have any experience with the grey mouse lemur as model organism, a somewhat more extensive description of relevant species specific information in relation to the housing, husbandry and procedures would be appreciated (see also some of the comments below).

We added supplementary information related to the grey mouse lemur in response to above and below Reviewer’s 1 comments (for example early development, nocturnality, reliance on olfactory cues... **lines 270-274, 398-402**) along with the already existing information about this species (**95-102**). We also added a reference gathering all needed information about the mouse lemur as a model for readers to would like to know more about it (**line 102**).

Reproduction:

17. The light regimes used during the study is unclear. Was there a gradual build up in length of light period to reach 14 hours and start breeding cycle? Were pregnant and lactating dams and their offspring also kept on the 14 hours light regime?

Lines 483-485: we added “The light regime transitions from short-day to long-day (and reciprocally) occur every six months abruptly, since photoperiodic phenotype change is more threshold-dependent and do not need gradual transition to happen”

Lines 503-505: we added “Since the long-day period lasts six months, mothers and their offspring were under the same light regime throughout the entire length of the study”

18. In case females were used for multiple breeding rounds: How many breeding cycles/year were initiated and what diets were provided in between breeding/ intervention periods?

Lines 494-499: we specified the needed information about the female which was used for two breeding rounds

Psychomotor tasks:

19. Is the grey mouse lemur a diurnal/ **nocturnal** species and what in what phase did behavior testing take place? Can you describe the test order on a testing day when multiple tests were performed? Was

test order of individuals randomized and was there habituation to the room in which testing took place?

Line 95: We specified that mouse lemurs are nocturnal.

lines 566-578: we added precisions about the timing of the testing, randomization and habituation. There was a habituation step for each task, but not a habituation to the room in itself, which would make the separation from the mother even longer than what it already was.

Line 594: we added that the horizontal rod task followed the negative geotaxis task on D+8

Line 634-635: we added that the visual discrimination task followed the rotarod© task on D+15 and D+30

20. Some tests use the nest box as motivation to escape to. Was this a neutral nest box or was it the individual's own nest (scent might be important). What happened to the mother and littermates during testing of individuals?

We used the individual's own nest, as we thought that the maternal scent might help increase the neonate motivation. The mouse lemurs indeed greatly rely on olfactory cues to communicate (although marking behavior such as urine washing only begin around D+40). To dispel any doubt, we specified where needed that it was the individual's own nest (**lines 596 and 639**)

Lines 573-576: we specified what happened to the mother and the littermates during the testing session "All tasks were performed in an adjacent room to the animal facility, while the mother remained in its cage. During a testing session, the animal's nest box was used as a positive reinforcement for the tested animal, while its littermates (if any) remained in a neutral nest box until their turn."

Statistical analysis:

21. Please explain how sample size was determined.

Please see our answer to point 15.

22. The mothers were subjected to the diet interventions during pregnancy and lactation, yet individual pups were used as the experimental units rather than mother/litter. Next to variation between dietgroups, It can be expected that there is some variation between mothers within diet groups as not all mothers/litters are exactly the same. For example there was variability in litter size as described in the results and litter size may by itself affect outcomes through various mechanisms as mentioned above. This calls for inclusion of litter of origin as random factor to the statistical models used. Same applies to the (diet) batch as there were multiple batches used. Please confirm that these factors were included or re-run the analysis with inclusion of these critical factors.

Litter size has been taken into account (see answers above) and, just as sex and weight, did not explain the performances and their removal lowered the AICc so we removed these factors from the models. However, we realized we did not take into account the mother identity into the model, and sincerely apologize for it. We ran again the tests after inclusion of mother identity as a random factor and updated the p-values and subsequent interpretations. Nevertheless, as you will see, it does not change the big picture. We updated the Statistical analysis section accordingly (**line 676-677 and 690**).

Figures

23. Figure 5 A: It is impossible to count the dots that represent the number of animals per group

succeeding and not succeeding in this figure as some of the dots are **overlapping**. Consider adapting the layout to allow people to discriminate individual dots. Alternatively, **add the actual numbers in the figure**.

Figure 5A (renamed Figure 4A): We made jittered scatterplot to limit the overlapping and add some transparency so even when slightly overlapped the dots could be distinguished. Moreover, we added the actual numbers.

Figure 4: Horizontal rod task results (D+8): **(a)** Logistic regression modelling the probability of success according to the distance from the nest box and the diet. Each animal, depicted by a point, either is successful – hence is assigned a value of 1 – or fails – hence is assigned a value of 0. The curve and its 95% confidence interval are built using a mixed generalized model using a binomial distribution. The VF curve differs significantly from the DF curve ($p=0.002$). **(b)** Box plot of the time to reach the nest box among successful animals for each distance. Number of successful animals for each distance level

and diet group is provided in Supplementary Table D. *: $p < 0.10$ (trend). Box plot elements: the centreline is the median, the box limits are the upper and lower quartiles, and the whiskers represent 1.5-fold the interquartile range.

24. **Figure 7 and 8:** see previous comment

Figure 7 (renamed Figure 6): we made jittered scatterplot as well, although we did not add the actual numbers since it would alter severely the readability of the plot

Figure 6: Negative binomial regression modelling the number of each rotating rod outcome per animal, i.e., falls (a), hanging on for 3 turns (b), jumps (c) and refusals (d), at D+15, D+22 and D+30 according to the diet. Each animal, depicted by a point, can be assigned a value from 0 to 5 out of 5 trials for each outcome. The curve and its 95% confidence interval are built using a mixed generalized model using a negative binomial distribution. The VF curve differed significantly from the DF curve only for 'falls' and 'hanging on for 3 turns' outcomes ($p=0.025$ and $p=0.046$, respectively). $N = 25$ in each diet group and for each trial.

Figure 8 (renamed Figure 7): since this plot contained much more dots of the same value, we replaced them by a bigger dot, the size of which is proportional to the number individual dots of this value. We added inside the dot the actual number of individual dots of this value.

Figure 7: Gamma regression modelling the time to find the exit across trials at D+15 and D+30 for the visual discrimination task. Each animal is depicted by a point. Animals that failed at this task are assigned the maximum value (300 s): the overlapping points inherited from these animals are melted in a bigger dot proportionally to their number which is displayed inside. The curves and their 95% confidence intervals are built using mixed generalized models using a Gamma distribution. The DF and VF curves did not differ significantly from one another ($p=0.436$ for D+15 and $p=0.305$ for D+30). $N = 25$ in each diet group and for each trial.

Concerning the comments made by reviewer 1:

This paper seeks to demonstrate dairy fat (DF) intake in mothers, pre- and post-birth while nursing, provides for improved psychomotor and cognitive performance in infant mouse lemurs than supplementing with vegetable-based fats (VF). Based on the data presented, the infant mouse lemurs did show greater neuromotor control and cognitive development based on visual discrimination tasks. While I was enthusiastic about this concept, I did not see a clear significance of this work presented. What is the reason for this work? It appears that the authors want to demonstrate the need for women to continue to use dairy products during pregnancy and nursing instead of relying on vegetable fats during this time. Although the abstract says: “Our results support further research on the impacts of dairy fat on cognitive performance after complete cessation of breastfeeding.” This was not presented in the manuscript. Throughout the Abstract, Introduction, and Discussion, I looked for a clear impact of this work, but did not find it.

One of my main concerns is in the experimental design. The study examines maternal supplement with either DF or VF but there is no control condition. What is the neural and cognitive performance of

infants without maternal supplement? How much did the DF or the VF improve normal neural and cognitive functioning? While the DF contained A and D vitamins, did the VF supplement? The studies relied upon 6 females during multiple pregnancies of litters (1,2, or 3 in a litter) for the subjects to test. Does this limit the power of the results? Many of the results exhibited large error bars making differences look less effective. What was the effect size for the analyses? This study could have been more impactful if it was written with a clear reason. Are the authors concerned that women will not using DF? Is there a concern that mothers or infants will only receive VF as in a vegan diet?

We thank Reviewer 2 for raising these concerns. However, we would like to clarify the experimental design, as we think there has been a misunderstanding arising from the word “supplement” used by Reviewer 2, and that we did not use in our study, on purpose. Indeed, this word suggests that we “added” dairy fat or vegetable fat in a standard diet (which would have been a control condition), which is not the case. Our lipidic input **is** the almost entire fat fraction of the diet (92.6%), which is already “balanced” regarding the proportion of calories originating from lipids (33%). Thus, providing a “control” condition would either mean:

- Removing DF or VF input, therefore providing a diet with only 5% of energy derived from lipids and more than 70% of energy from carbohydrates which is extremely unbalanced
- Or providing another lipidic input, in the same proportions as DF and VF, but modifying the proportion of each fatty acid (as we already did with DF and VF diets design), bringing infinite possibilities of food sources to use (fish oil? olive oil? Red meat fat?)

This is the reason we consider DF and VF are control conditions one from another, since they differ in the quality of fatty acids provided. This point of view is supported by two additional arguments:

- The lipidic fraction of the DF diet consists of 75% butter and 25% vegetable oils, meaning that the DF diet is actually a modification of the VF diet itself, by decreasing the part originating from vegetable oil and adding lipids from dairy products
- The VF diet is a representation, in some way, of the western diet, which brings concerns arising from the high palmitic and linoleic acids intake. Our DF diet is an attempt to correct this deleterious fatty acid profile, by proposing an alternative source of animal fatty acids instead of fish and red meat that have been associated with health concerns (either heavy metal contamination for the first one, or colorectal cancers and cardiovascular diseases for the later).

Therefore, we acknowledge that the rationale wasn't clear and strong enough and improved our introduction and discussion in order to 1) better underline the potential impact and significance of our work, and 2) better justify the relevance of the comparison between DF and VF diets. We also ask the reviewers and the editors if introducing the VF diet as a control diet depicting the western diet in the introduction would help the readers understand the experimental design. You will see these modifications **lines 40-55, 83-92, 102-110, 426-433 and 452-454.**

For more specific remarks made by Reviewer 2:

- **What is the neural and cognitive performance of infants without maternal supplement? How much did the DF or the VF improve normal neural and cognitive functioning?**
 - ⇒ The performances during early development have only been studied once (Boulinguez-Ambroise, 2020). We already cited this study to compare with our results for the rotarod© task (**lines 283-297**). This study also assessed the performance in grasping strength and locomotor posture, which we did not assess in our study, but we still used these valuable

results for discussion (**lines 231-236** and **lines 263-266**). There is no other data about neonatal performances (although we added information related the developmental psychomotor trajectories, as also asked by Reviewer 1). Thus, we believe we could not address this question more than we already did

- **While the DF contained A and D vitamins, did the VF supplement?**
 - ⇒ We did not analyze the experimental diets for A and D vitamins, we only specified in the introduction that dairy fat usually also contain these vitamins (and that we could not exclude that the observed effects might originate from other nutrients than the fatty acids). Thus, we do not have the information about the A and D vitamin concentrations here, neither for both diets. However, the oils used (olive, rapeseed, palm and sunflower) are known to contain neither of these vitamins, although they contain the precursor of vitamin A (carotenoids).
- **The studies relied upon 6 females during multiple pregnancies of litters (1,2, or 3 in a litter) for the subjects to test. Does this limit the power of the results?**
 - ⇒ There was indeed a female that gave birth twice, resulting in 6 pups. It is true we did not take into account the maternal origin at first in our models. We sincerely apologize for this error and, as asked by both Reviewers, we added the mother as a random factor in our models. We updated the results accordingly, although it did not change substantially the observed effects.

REVIEWERS' COMMENTS:

Reviewer #1 (Remarks to the Author):

I thank the authors for addressing all questions raised and for implementing the changes to the manuscript. I confirm that all previous comments have been appropriately addressed.

Reviewer #2 (Remarks to the Author):

No further comments.